



# Radium-228-derived ocean mixing and trace element inputs in the South Atlantic

Yu-Te Hsieh[1], Walter Geibert[2], E. Malcolm S. Woodward[3], Neil J. Wyatt[4], Maeve C. Lohan[4], Eric P. Achterberg[4,5], Gideon M. Henderson[1]

[1]Department of Earth Sciences, University of Oxford, UK
[2]Alfred Wegener Institute Helmholtz Centre for Polar and Marine Research, Bremerhaven, Germany
[3]Plymouth Marine Laboratory, Plymouth, UK
[4]Ocean and Earth Sciences, National Oceanography Centre, Southampton, UK
[5]GEOMAR Helmholtz Centre for Ocean Research, Kiel, Germany

*Correspondence to*: Yu-Te Hsieh (yu-te.hsieh@earth.ox.ac.uk)

**Abstract.** Trace elements play important roles as micronutrients in modulating marine productivity in the global ocean. The South Atlantic around 40ºS is a prominent region of high productivity and a transition zone between the nitrate-depleted

Subtropical Gyre and the iron-limited Southern Ocean. However, the sources and fluxes of trace elements to this region remain unclear. In this study, the distribution of the naturally occurring radioisotope $^{228}$Ra in the water column of the South Atlantic (Cape Basin and Argentine Basin) has been investigated along a 40ºS zonal transect to estimate ocean mixing and trace element supply to the surface ocean. Ra-228 profiles have been used to determine the horizontal and vertical mixing rates in the near-surface open ocean. In the Argentine Basin, horizontal mixing from the continental shelf to the open ocean

shows an eddy diffusion of $K_x = 1.7 \pm 1.4$ ($10^6$ cm$^2$ s$^{-1}$) and an integrated advection velocity $w = 0.6 \pm 0.3$ cm s$^{-1}$. In the Cape Basin, horizontal mixing is $K_x = 2.7 \pm 0.8$ ($10^7$ cm$^2$ s$^{-1}$) and vertical mixing $K_z = 1.0 - 1.5$ cm$^2$ s$^{-1}$ in the upper 600 m layer. Three different approaches ($^{228}$Ra-diffusion, $^{228}$Ra-advection and $^{228}$Ra/TE-ratio) have been applied to estimate the dissolved trace-element fluxes from shelf to open ocean. These approaches bracket the possible range of off-shelf fluxes from the Argentine margin to be: 3.8 – 22 ($\times 10^3$) nmol Co m$^{-2}$ d$^{-1}$,  7.9 – 20 ($\times 10^4$) nmol Fe m$^{-2}$ d$^{-1}$ and 2.7 – 6.5 ($\times 10^4$) nmol Zn m$^{-2}$ d$^{-1}$

Off-shelf fluxes from the Cape margin are: 4.3 – 6.2 ($\times 10^3$) nmol Co m$^{-2}$ d$^{-1}$,  1.2 – 3.1 ($\times 10^4$) nmol Fe m$^{-2}$ d$^{-1}$ and 0.9 – 1.2 ($\times 10^4$) nmol Zn m$^{-2}$ d$^{-1}$. On average, at 40ºS in the Atlantic, vertical mixing supplies 0.4 – 1.2 nmol Co m$^{-2}$ d$^{-1}$, 3.6 – 11 nmol Fe m$^{-2}$ d$^{-1}$, and 13 – 16 nmol Zn m$^{-2}$ d$^{-1}$ to the euphotic zone. Compared with atmospheric dust and continental shelf inputs, vertical mixing is a more important source for supplying dissolved trace elements to the surface 40ºS Atlantic. It is insufficient, however, to provide the trace elements removed by biological uptake. Other inputs (e.g. particulate, or from

winter deep-mixing) are required to balance the trace element budgets in this region.



# 1 Introduction

While trace elements (TEs) play important roles as micronutrients for marine productivity in the surface ocean (Morel and Price, 2003; Lohan and Tagliabue, 2018), their sources and fluxes to the ocean remain poorly constrained. For example, the South Subtropical Convergence (SSTC) in the South Atlantic, near 40°S, is a prominent high productivity region and an important zone for the dynamic inter-ocean exchange between the Atlantic, Pacific, Indian, and Southern Oceans (Fig. 1). This region is also a transition zone between the nitrate-depleted Subtropical Gyre and the iron-limited Southern Ocean, creating one of the most dynamic nutrient environments in the world oceans (Moore et al., 2004). The chlorophyll concentration is generally high (0.2-0.3 mg chlorophyll a m$^{-3}$) in this region (Longhurst, 2007). Modeling and experimental studies have both suggested that this region is iron limited or co-limited (Moore et al., 2004; Browning et al., 2014; 2017). However, the sources and fluxes of TEs to this region are not well quantified.

Previous work has assessed TE input to the wider South Atlantic from rivers (Vieira et al., 2020), atmospheric dust (Gaiero et al., 2013), shelf sediments (Graham et al., 2015), the Agulhas current (Paul et al., 2015) and hydrothermal vents (Saito et al., 2013). There are also studies of TE distributions in some areas of the South Atlantic (e.g. Chever et al., 2010; Bown et al., 2011). Two UK-GEOTRACES cruises in 2010-2012 provided a significant increase to such observations, focusing particularly on 40°S (Homoky et al., 2013; Browning et al., 2014; Wyatt et al., 2014, 2020; Chance et al., 2015; Menzel Barraqueta et al., 2019). These studies did not assess the fluxes of TEs by ocean mixing and transport. In this study, we address this issue using samples taken on the UK-GEOTRACES cruises.

Oceanic mixing and advection play an important role in transporting nutrients to the euphotic zone (Oschlies, 2002). The distribution of TEs in the surface ocean is primarily controlled by the inputs from the continental shelves (i.e. rivers, submarine groundwater discharge (SGD) and sediments), deep ocean waters (regeneration, continental slopes, and hydrothermal vents) and aeolian inputs, with these mediated by lateral and vertical mixing (diffusive/turbulent mixing), advection, particle scavenging and biological uptake. In particular, deep winter mixing has been shown to be an important mechanism bringing TEs from below the mixed layer to the surface ocean (Tagliabue et al, 2014; Achterberg et al., 2018; 2020; Rigby et al., 2020). Geochemical tracers for ocean mixing can therefore be used to indirectly estimate TE inputs and outputs in the upper ocean, e.g. tritium-$^3$He (Jenkins, 1988; Schlitzer, 2016) and radium isotopes-$^{228}$Ra (Cai et al., 2002; Ku et al., 1995; Nozaki and Yamamoto, 2001; Sarmiento et al., 1990; Moore, 2000; Charette et al., 2007; Sanial et al. 2018).

The four naturally occurring radium isotopes cover a wide range of half-lives ($^{226}$Ra, $T_{1/2}$ = 1600 years; $^{228}$Ra, $T_{1/2}$ = 5.75 years; $^{223}$Ra, $T_{1/2}$ = 11.4 days; $^{224}$Ra, $T_{1/2}$ = 3.66 days), which enables us to study oceanic processes at different time scales. Ra-228 is continuously produced through the decay of $^{232}$Th in shelf sediments, released into seawater, and then transported into the surface open ocean by mixing or advection. The half-life of $^{228}$Ra is much shorter than the estimated Ra residence time by removal of ~500 years (Moore and Dymond, 1991). The distribution of $^{228}$Ra in the ocean is therefore mainly controlled by ocean transport and radioactive decay, and can be used to estimate lateral mixing from the coastal shelf or





continental slope to the open ocean (Kaufman et al., 1973; Knauss et al., 1978; Yamada and Nozaki, 1986; Sanial et al. 2018). Subsequent downward mixing from the surface can also be used to assess vertical mixing in the upper water column

(Charette et al., 2007; Sarmiento et al., 1976; van Beek et al., 2008). Radium-228 has also been used as a conservative tracer to estimate submarine groundwater discharge (SGD) (Windom et al., 2006; Moore et al., 2008; Kwon et al., 2014; Rodellas et al., 2015; Le Gland et al., 2017), river inputs (Vieira et al., 2020), continental shelf (van der Loeff et al., 1995; Charette et al., 2016; Sanial et al. 2018; Kipp et al., 2018a) and hydrothermal inputs (Kipp et al., 2018b).

In this study, we investigate the distributions of $^{228}$Ra, as well as $^{226}$Ra, in both the Argentine and Cape Basins of a 40ºS

latitudinal transect in the Atlantic Ocean (Fig. 1). This is also the first exploration of the vertical and horizontal $^{228}$Ra distributions reported for the Cape Basin. We investigate the application of seawater $^{228}$Ra as a tracer for vertical and horizontal mixing in the surface South Atlantic, to provide estimates of the dissolved TE fluxes, with a focus on cobalt, iron and zinc, in the micronutrient-depleted euphotic zone.

## 2 Methods

### 2.1 Study sites and sampling

Seawater samples were collected from 14 stations on the RRS *Discovery* and RRS *James Cook* during two UK GEOTRACES cruises, GA10E (D357) and GA10W (JC068), along the 40ºS transect of the Atlantic Ocean (Fig. 1). The first cruise (D357) took place in the SE Atlantic (Cape Basin) between October and November 2010; the second cruise (JC068) took place along the whole 40ºS transect between December 2011 and January 2012. For radium isotope analyses, the Cape

Basin samples were only taken from D357, and the Argentine Basin samples from JC068. Stations from these cruises are shown, along with those from previous Ra studies (e.g. GEOSECS and the Transient Tracers in the Ocean, TTO) in this region (Fig. 1a).

The surface currents show dynamic interaction and mixing on both the western and eastern sides of the Atlantic basins at 40ºS. In the Argentine Basin, the Rio de la Plata estuary is located on the western margin of the transect. The boundary

currents Brazil Current (BrC) and Malvinas Current (MC) meet between 33ºS and 45ºS along the continental margin of South America and these become the South Atlantic Current (SAC) transporting water eastwards along 40ºS. The water from the BrC is captured between Stn20 and Stn21 with a strong west to east gradient in salinity and other chemical and physical properties, which also suggests a limited exchange of water across the continental shelf break (see below). In the Cape Basin, the SAC turns northeast before reaching the continent of Africa, and the Agulhas Current (AC) adds warm water eddies from

the Indian Ocean. The warm and salty water of the AC was sampled in the top 500 m at Stn2 (Fig. 1b). These currents meet and become the Benguela Current (BeC) flowing through the Cape Basin and into the South Atlantic.

A total of 48 samples were analysed for $^{228}$Ra/$^{226}$Ra ratio and $^{228}$Ra concentration, and 33 samples for $^{226}$Ra concentration, collected using three different sampling techniques during the cruise. Surface seawater samples (80 – 100 L) at 5 m depth



were collected via a trace-metal clean seawater supply (fish) using a Teflon bellow pump (Almatec-A15) and acid-cleaned
tubing. Samples between 50 m and 400 m were collected using a standard CTD rosette, typically sampling from four 20 litre
Niskin bottles. These samples were stored briefly in low-density polyethylene (LDPE) cubitainers and then filtered through
Mn-fibre cartridges by gravity (flow rate < 0.5 L min$^{-1}$) on-board for the extraction of radium isotopes (Moore et al., 1985;
Reid et al., 1979). In addition, large-volume seawater sampling (300 – 600 L) was carried out using in-situ stand-alone pump
systems (SAPs), pumping sea water over Mn-fibres in polypropylene cartridges (van Beek et al., 2008) at flow rate of 2-5 L
min$^{-1}$ at three selected sampling stations (Stn1, Stn3 and Stn4.5). All samples collected by these collection techniques (pump,
CTD, SAP) were used for measurement of $^{228}$Ra/$^{226}$Ra ratios by mass spectrometry, and $^{224}$Ra, $^{223}$Ra, $^{228}$Th, and $^{227}$Ac
activities using a delayed coincidence RaDeCC device. For most samples, a separate sample of 250 mL seawater was also
collected for measurement of $^{226}$Ra concentration by mass spectrometry.

Trace element (TE) samples were collected using a titanium CTD rosette fitted with trace metal clean Teflon-coated Niskin
bottles and filtered on-board through 0.8/0.2 μm polyethersulfone (PES) membrane cartridge filters (AcroPak500$^{TM}$, Pall),
before analysis. All the TE data and fluxes reported and discussed in this study refer to the dissolved fraction only. The data
of zinc and cobalt were measured and published by Wyatt et al. (2014 and 2020). Some of the iron data were determined and
published by Browning et al. (2014) and Clough et al. (2016). All the TE data are available on the GEOTRACES
Intermediate Data Product (IDP) 2017 (Schlitzer et al., 2018).

**2.2 Ra isotopes analysis**

Mn-fibre samples were first counted on-board using a 4-channel radium delayed coincidence counting system (RaDeCC) for
$^{224}$Ra, $^{223}$Ra, $^{228}$Th and $^{227}$Ac (Moore and Arnold, 1996), but the data for $^{224}$Ra, $^{223}$Ra, $^{228}$Th, and $^{227}$Ac are not discussed in
this paper. After the counting, Ra was purified using the following procedure for precisely measuring $^{228}$Ra/$^{226}$Ra ratios and
$^{228}$Ra concentrations by MC-ICP-MS (Hsieh and Henderson, 2011). Mn-fibres were ashed at 550°C for 6 hours and then
leached with distilled 6N HCl to remove Ra from the ashed fibres. Ra was then co-precipitated with Sr(Ra)SO$_4$ in the
leached solution, centrifuged and cleaned with 3N HCl and pure H$_2$O a few times until the pH was > 4. To convert
Sr(Ra)SO$_4$ to Sr(Ra)CO$_3$, 2mL 1M Na$_2$CO$_3$ solution was added and heated on a hotplate for 3 hours. After centrifuging and
discarding the supernatant, Sr(Ra)CO$_3$ was finally dissolved in 2 ml 6N HCl for ion exchange column chemistry, using
AG50-X8  (to separate Ra and Ba from $^{228}$Th and other matrix elements, e.g. Ca, Sr and Mn) and Sr-Spec (to separate Ra
from Ba to avoid molecular interferences during MC-ICP-MS analysis).

Smaller seawater samples (250 mL) collected for $^{226}$Ra were spiked with a $^{228}$Ra-spike (Hsieh and Henderson, 2011) and the
Ra was purified by the precipitation of CaCO$_3$ and processing with ion exchange columns of AG1-X8, AG50-X8 and Sr-
Spec for the measurement of $^{226}$Ra concentrations by MC-ICP-MS (Foster et al., 2004). Overall chemical blanks were
monitored throughout the whole chemical procedures, and were found to contribute less than 1% of the $^{226}$Ra in the sample
and were not detectable for $^{228}$Ra.



Measurements of [228]Ra and [226]Ra were performed using a Nu Instrument MC-ICP-MS using the protocols described in Hsieh and Henderson (2011). Ra-228 and [226]Ra were measured simultaneously on two ion counters. The uranium standard CRM-145, was used to bracket each sample for the mass bias and ion counter gain corrections. Instrumental memories of [228]Ra and [226]Ra were also detected on ion counters before each measurement. The machine memory was about $0.2 \pm 0.1$ cps

(counts per second) (n = 16, 2S.E.) The memory correction was insignificant for [226]Ra, because the ratio of memory to sample signal is small ($< 10^{-4}$). However, the memory correction could be significant for samples with low [228]Ra activities and count rates. For instance, the count-rate during analysis of [228]Ra on the sample collected at 4741 m at Stn4.5 was only 0.5 cps. At this low count-rate, instrumental memory contributed ~40% of the signal to the sample [228]Ra signal and the uncertainty of memory correction becomes substantial. In this study, most of the surface and deep waters in the South

Atlantic were measured at count rates > 2 cps [228]Ra, which provides assurance that the contribution of the instrumental memory uncertainty to the total uncertainty is < 10%.

For samples without accompanied [226]Ra measurements, silica data (Table 1) are used to assess [226]Ra activities (Appendix A). The global ocean [226]Ra-Si relationship from the [226]Ra and silica data from the GEOSECS and TTO datasets (Fig. A1) is used to determine [226]Ra activities in the 27 cases where no subsamples were collected for separate [226]Ra analysis (such [226]Ra

estimates are shown in brackets in Table 1). The relationship has a slope of 0.143 dpm 100L$^{-1}$ of [226]Ra per µmol L$^{-1}$ of Si and an intercept of 8.0 dpm 100 L$^{-1}$, which is comparable with the average slope of 0.1 observed by Broecker et al. (1976) in the Atlantic. The Si-extrapolated [226]Ra and measured [226]Ra activities (data from this study and the TTO) show a relatively consistent result although the extrapolated [226]Ra has a larger uncertainty ($\pm11\%$ 2S.E.) than the measured [226]Ra uncertainty ($\pm4\%$, 2S.E.) The uncertainties of [226]Ra activity have been used in the error propagation of [228]Ra activity; the total

uncertainty of [228]Ra is typically about 6 – 12% (2S.E.)

### 2.3 [228]Ra-derived 1-D mixing models

The distribution of seawater [228]Ra in the ocean is mainly controlled by mixing, advection, radioactive decay and additional removal/input. It has been widely used as a tracer for measuring diffusion coefficients and advection rates on a basin-wide scale in the surface or at intermediate depths in the ocean (e.g. Cochran, 1992; Ku and Luo, 2008; Sanial et al., 2018). The

one-dimensional (1-D) [228]Ra advection-diffusion model is commonly expressed by the formula (e.g. Moore, 2015):

$$\frac{\partial A}{\partial t} = K_x \frac{\partial^2 A}{\partial x^2} - w \frac{\partial A}{\partial x} - \lambda A \pm J \qquad (1)$$

where A is activity of [228]Ra, t is time, $K_x$ is horizontal eddy diffusion coefficient, w is advection velocity, x is offshore distance, λ is decay constant ($\lambda_{Ra228} = 3.82 \times 10^{-9}$ s$^{-1}$), and J is additional input or removal of [228]Ra.

To use the model to accurately calculate the mixing rates, several assumptions need to be made:

*(1) Steady state (∂A/∂t = 0)*





This assumption requires long-term monitoring of [228]Ra activities in the ocean due to the long half-life of [228]Ra. Charette et al. (2015) compared the [228]Ra data in the North Atlantic from the US GEOTRACES and the TTO programs, and found that the upper ocean [228]Ra inventories have remained constant over the past 30 years. Although [228]Ra seasonality in coastal areas may introduce uncertainty to the mixing model, the assumption of steady state is likely to be valid for [228]Ra on the decade

timescales and at ocean-basin scales. The comparison between our [228]Ra data and the limited data from the TTO in this region show good agreement (Fig. 2b), supporting this assumption for the South Atlantic.

*(2) No additional input or removal of [228]Ra ($\pm J = 0$)*

Based on particle removal, Ra residence time is estimated to be ~500 years in the surface ocean (Moore and Dymond, 1991). However, there is no measurable particle removal of Ra in the surface open ocean at the time scale of [228]Ra half-life (5.75

years) (Moore, 2015). Vertical mixing could affect the horizontal distribution of [228]Ra in the surface water, which would require 2-D models to resolve the problem. However, the sample resolution is not good enough for precise 2-D modeling in this study. Therefore, we only apply the 1-D model to estimate the maximum horizontal mixing rates by neglecting the term of downward mixing.

*(3) Boundary conditions ($A = A_0$ at $x = 0$ and $A = 0$ at $x \to \infty$)*

The boundary condition of $A = 0$ at $x \to \infty$ is incorrect for using [228]Ra to determine coastal mixing as the distribution of [228]Ra could be controlled by water mass mixing within the coastal distance scale (< 50 km) rather than eddy diffusion (Moore, 2000). Although, on the ocean-basin scale, this boundary condition may be valid, as the major sink of [228]Ra in the ocean is radioactive decay, observed seawater [228]Ra is still not completely zero in the remote ocean. To avoid this problem, we follow the suggestion of Moore (2015) and define the [228]Ra excess ([228]Ra$_{ex}$ = [228]Ra − [228]Ra$_{bg}$) by subtracting the

background value in the middle of the South Atlantic, [228]Ra$_{bg}$: $0.23 \pm 0.06$ dpm 100 L$^{-1}$ (1SE, n=6). This value is determined by the average of the observed values at the water depth between 1000 and 3500 m (except 2580 m at Stn1 on the continental slope) in this study and a previous study around the 40°S transect (Hanfland, 2002; Fig.1, ANT XV/4 station S8, S10 and S11). This mid-water [228]Ra background cannot be supported by in-situ production from [232]Th in the water column, because the activity of [232]Th is ≈100 times lower than that of [228]Ra  ([232]Th = 1~20 ×10$^{-4}$ dpm 100 L$^{-1}$, Deng et al., 2014). In

this case, the boundary condition can be rewritten as $A_{ex\_0} = A_0 − A_{bg}$ at $x = 0$ and $A_{ex} = 0$ at $x = \infty$. Considering the assumptions discussed above, Eqn. (1) can be written as:

$$0 = K_x \frac{\partial^2 A_{ex}}{\partial x^2} - w \frac{\partial A_{ex}}{\partial x} - \lambda A_{ex} \qquad (2)$$

In this study, we consider two scenarios in the horizontal [228]Ra calculations: (1) mixing only ($w = 0$); and (2) advection only ($K_x = 0$). We use these two scenarios to provide independent assessments of chemical fluxes in the surface ocean, to bracket

the range of possible TE fluxes that are consistent with the [228]Ra data regardless of the combination of mixing and advection





in the real ocean. Using the boundary conditions, $A_{ex\_0} = A_0 - A_{bg}$ at $x = 0$ and $A_{ex} = 0$ at $x = \infty$, Eqn. (2) can be solved for diffusive mixing only:

$$A_{ex} = A_{ex\_0} \exp(-ax), where\ a = \sqrt{\lambda/K_x} \tag{3}$$

and for advection only:

$\qquad w = \lambda x/\ln\left(A_{ex\_0}/A_{ex}\right) = X_{1/2}/T_{1/2} \tag{4}$

where $A_{ex\_0}$ is the activity of $^{228}Ra_{ex}$ at $x = 0$ (i.e. Stn0 or Stn25); $X_{1/2}$ is the distance at which $A_{ex} = 0.5A_{ex\_0}$; and $T_{1/2}$ is the half-life of $^{228}Ra$ (5.75 years). The exponential fit of the surface $^{228}Ra$ data provides the estimate of the maximum diffusion coefficients ($K_x$) at both ends of the transect, and the linear fit of the surface $^{228}Ra$ data after the shelf break in the Argentine basin provides the minimum estimate of the advection water transport ($w$) along the west end of the transect (see discussion

below).

### 3 Results

#### 3.1 Ra isotope concentrations

The results of $^{226}Ra$ and $^{228}Ra$ activities and the activity ratios of $^{228}Ra/^{226}Ra$ are presented in Table 1 (with $\pm$ 2S.E.) The vertical profiles of measured $^{226}Ra$ shows good agreement with the data from the closest GEOSECS and TTO stations in this

region (e.g. Ku and Lin, 1976; Key et al., 1990; 1992a,b) (Fig. 2a). Ra-226 activities range from 8.2 to 22.4 dpm 100 L$^{-1}$. In this study, the results of $^{226}Ra$ are mainly used for calculating $^{228}Ra$ activities, and will not be discussed in further detail.

The activity ratios of $^{228}Ra/^{226}Ra$ range from 0.017 to 1.599 in the surface water and are comparable with the ratios from 0.080 to 2.810 observed in previous studies in this region (TTO data, Windom et al., 2006 and Hanfland, 2002). The vertical profiles of $^{228}Ra$ activity are shown in Fig. 2b. In surface waters, the activities of $^{228}Ra$ ranging from 1.02 to 17.66 dpm 100

L$^{-1}$ in the Argentine Basin are consistent with the observed values from 0.07 to 24.0 dpm 100 L$^{-1}$ from the previous studies (TTO data and Windom et al. 2006), and the activities of $^{228}Ra$ ranging from 0.99 to 3.22 dpm 100 L$^{-1}$ in the Cape Basin are consistent with the observed values from 0.67 to 4.23 dpm 100 L$^{-1}$ from the previous study (Hanfland, 2002). Between 600 m and 4000 m, $^{228}Ra/^{226}Ra$ ratios decrease to 0.015 – 0.030 and $^{228}Ra$ activities decrease to 0.29 – 0.32 dpm 100 L$^{-1}$. In the 100 m closest to the ocean floor, the ratios of $^{228}Ra/^{226}Ra$ increase to 0.048 – 0.088 and the activities of $^{228}Ra$ also increase to

1.07 – 1.94 dpm 100 L$^{-1}$. This is the first dataset of seawater $^{228}Ra$ reported in the intermediate and deep waters in the Cape Basin. It shows higher $^{228}Ra$ values than those observed in intermediate ($< 0.1$ dpm 100 L$^{-1}$) and deep waters (0.22 – 0.28 dpm 100 L$^{-1}$) to the south in the Southern Ocean (Charette et al., 2007; van Beek et al., 2008). The vertical profiles of seawater $^{228}Ra$ in the Cape Basin are consistent, however, with expectations based on GEOSECS and TTO observations elsewhere in the Atlantic Ocean (Moore et al., 1985).



### 3.2 Micronutrient concentrations

Dissolved Co, Fe and Zn concentration data (Wyatt et al., 2014 and 2020; Browning et al., 2014; Clough et al., 2016; Schlitzer et al., 2018) are summarised in Appendix B (Table B1 and B2). At the continental margins, surface shelf waters show much higher trace element concentrations (Co: 146.2 pM, Fe: 1.53 nM, and Zn: 0.59 nM in the Argentine margin; Co: 46.9 pM, Fe: 0.35 nM, and Zn: 0.14 nM in the Cape margin) than observed in the open ocean surface waters along the 40°S transect (Fig. 3). These trace element concentrations are generally low in the surface mixed layer and increase with depth below the mixed layer (Fig. 4). In the upper 500 m, Co concentrations range from 1.6 to 61.2 pM, and Fe and Zn concentrations range from 0.05 to 0.54 and 0.01 to 0.53 nM, respectively.

## 4 Discussion

### 4.1 $^{228}$Ra-derived horizontal mixing and advection

#### 4.1.1 Argentine Basin

The distribution of $^{228}$Ra$_{ex}$ in the Argentine Basin (GA10W) is controlled by the Rio de la Plata river plume and the Brazil Current (Fig. 5a), and this is supported by a good correlation with salinity (linear regression $R^2 = 0.96$; Fig. C1 in Appendix C). If we apply the 1-D mixing model (Eqn.3) to fit the $^{228}$Ra$_{ex}$ data between Stn25 and Stn21, across the boundary of the Brazil Current, the estimate of the offshore horizontal diffusion coefficient $K_x$ is $1.7 \pm 1.4 \times 10^6$ cm$^2$ s$^{-1}$, which is likely to be an overestimate due to the influence of river plume and the advection of boundary current. Nevertheless, this estimate is still within the range of other estimates of $K_x$ between $10^5$ and $10^8$ cm$^2$ s$^{-1}$ in a variety of margin and open ocean settings (e.g. Kaufman et al., 1973; Knauss et al., 1978; Yamada and Nozaki, 1986).

Across the boundary of the Brazil Current, the advection of eastward flowing SAC shows a significant impact on the distribution of $^{228}$Ra$_{ex}$ in the surface Argentine Basin (Fig. 5a). If we apply the 1-D advection model (Eqn.4) to fit the data between Stn21 and Stn18 (Fig. 5a), the $X_{1/2}$ is $1151 \pm 613$ km (from Stn 21) and the estimate of average advection velocity $w$ is $0.6 \pm 0.3$ cm s$^{-1}$. Although the $^{228}$Ra-derived velocity is smaller than the typical velocities (2 ~ 4 cm s$^{-1}$) around the South Atlantic Subtropical Gyre (Schlitzer, 1996), similar advective $^{228}$Ra signals have been previously observed in other surface ocean current systems, including the Peru and Kuroshio currents in the Pacific (Knauss et al. 1978 Yamada and Nozaki 1986).

#### 4.1.2 Cape Basin

Ra-228 data from the Cape transect (GA10E) are used to calculate the offshore horizontal diffusion coefficient ($K_x$) in the Cape Basin (Fig. 5b). Applying the simple 1-D mixing model, assuming w = 0, to fit the surface $^{228}$Ra$_{ex}$ data in the Cape Basin (Fig. 5b), the estimate of $K_x$ is $2.7 \pm 0.8 \times 10^7$ cm$^2$ s$^{-1}$, which is an order of magnitude higher than the value observed in the Argentine Basin (see above), but within the range of other observed values in the oceans ($10^5$ ~ $10^8$ cm$^2$ s$^{-1}$).





The horizontal diffusion coefficient $K_x$ is likely to be overestimated in the Cape Basin due to the influence of episodic water advection. The Agulhas Current Leakage (ACL) is known for transporting water from the Indian Ocean into the South Atlantic and episodically introduces eddies (Agulhas Rings) into the Cape Basin (Beal et al., 2011). However, the signals of mixing and advection cannot be easily separated with the $^{228}$Ra data alone (Fig. 5b). For example, the distribution of $^{228}$Ra$_{ex}$ in the surface Cape Basin shows elevated values (Stn2 and Stn4.5) above the fitted curve and coincides with the elevated

salinity and temperature data (Fig. 5d), which indicates that the elevated $^{228}$Ra$_{ex}$ is likely to come from an advective signal (e.g. ACL). The ACL signal has also been identified with a distinct Pb isotope signature in the upper water column at Stn2 (Paul et al., 2015). The application of a 1-D mixing model may actually be biased by the addition of these high $^{228}$Ra$_{ex}$ waters, therefore, the horizontal diffusion coefficient $K_x$ is likely to be at its maximum estimate for the Cape Basin. Nevertheless, the overall gradient of $^{228}$Ra$_{ex}$, decreasing along the distance away from the shore, is driven by the loss of $^{228}$Ra

through both water mixing and radioactive decay.

Despite uncertainty in the diffusion coefficients due to advection of other sources, the $^{228}$Ra data do place bounds on maximum horizontal mixing in the surface ocean away from the eastern and western boundaries of the Atlantic at 40°S. These bounds can be used to quantify the trace element inputs from the continental margins to the South Atlantic (see Discussion 4.3).

**4.2 $^{228}$Ra-derived vertical mixing**

The calculation of vertical (diapycnal) mixing ($K_z$) is based on a situation in which $^{228}$Ra is mixed horizontally away from the coast in the surface mixed layer and then down into the subsurface. The 1-D mixing model (Eqn.3) can therefore be applied to the depth profiles of $^{228}$Ra$_{ex}$ in the upper 600 m of the water columns to calculate $K_z$ near the surface ocean (Fig. 6). In the 1-D mixing model, the term of diapycnal advection is generally negligible, as the oceanic vertical advection velocity is usually very small, i.e. $10^{-3} \sim 10^{-5}$ cm s$^{-1}$ (Liang et al., 2017). The best-fit exponential curves in $^{228}$Ra$_{ex}$ activities to depth (z)

below the surface mixed layer are used in the same 1-D mixing model (Eqn.3) to calculate vertical mixing $K_z$ for the upper ≈600 m of each station in both the Argentine and Cape Basins (Fig. 6).

Vertical diffusion coefficients ($K_z$) are calculated at six stations where the depth profiles of $^{228}$Ra$_{ex}$ are available (Stn1, Stn2, Stn3, Stn4.5, Stn18 and Stn21), and resulting $K_z$ values range from 1 to 65 cm$^2$ s$^{-1}$ at these stations (Fig. 6). The high K$_z$

values of 65 cm$^2$ s$^{-1}$ and 7 cm$^2$ s$^{-1}$ at Stn18 and Stn21, respectively, are most likely biased by the lateral inputs of $^{228}$Ra below the mixed layer (see later discussion). Excluding the values of Stn18 and Stn21, the range of $K_z$ values from 1.0 to 1.5 cm$^2$ s$^{-1}$ is broadly comparable to the average $K_z$ of 1.5 cm$^2$ s$^{-1}$ assessed from tritium measurements in the South Atlantic (Li et al., 1984). These estimates are also consistent with the range of observed vertical mixing from 0.1 to 10 cm$^2$ s$^{-1}$ using different methods (e.g. $^7$Be, SF$_6$ dye release and microstructure shear probe methods) in different ocean-basin settings (Kunze and

Sanford, 1996; Ledwell et al., 1993; Martin et al., 2010; Painter et al., 2014; Kadko et al., 2020).





Given that the calculation of $K_z$ is based on the vertical gradients of $^{228}$Ra driven by vertical mixing and radioactive decay only, it therefore relies on the assumption that the vertical gradients are not dominated by lateral input of $^{228}$Ra at depths below the surface mixed layer. This assumption is supported by an inspection of horizontal $^{228}$Ra gradients at depths below the mixed layer (Fig. 7). Due to the sample resolution, detailed inspection is only available for the Cape Basin. Here, unlike

the exponential change seen in the surface layer, $^{228}$Ra$_{ex}$ activities at 50 m and deeper do not show an increasing gradient towards the continental margin. This therefore argues against lateral mixing away from the shore as the major mechanism driving sub-surface $^{228}$Ra concentrations in the Cape Basin. In addition, the distribution of $^{228}$Ra on an isopycnal surface (sigma-t of ~ 26.7 kg m$^{-3}$; ≈200 m depth) is largely constant and shows no lateral gradient (Fig. 7). Studies using theoretical models to simulate seawater $^{228}$Ra distribution have also shown that the horizontal eddy mixing ($K_x$) has little effect on the

vertical distribution of $^{228}$Ra (Lamontagne and Webster, 2019).

The depth profiles of $^{228}$Ra$_{ex}$ in the Argentine Basin show evidence of advective $^{228}$Ra below the mixed layer, potentially from the nearer shore shelf waters. For example, elevated $^{228}$Ra$_{ex}$ values are seen around 400 m at Stn18 and 200 m at Stn21 (Fig. 6a and b) and these may explain the extremely high $K_z$ values at these stations. Although the possibility of lateral inputs cannot be entirely excluded, particularly in the Argentine Basin, the vertical variation of $^{228}$Ra near the surface mixed layer

still provides the first estimates of maximum vertical mixing and the upper limits of trace element inputs from vertical mixing to the surface ocean along the 40ºS transect.

### 4.3 Trace element inputs in the South Atlantic

Trace elements (TEs) are important micronutrients for marine productivity in the surface ocean. However, the inputs of these TEs to the euphotic zone in the South Atlantic are still unknown. In this study, we consider three different $^{228}$Ra approaches

to quantify the horizontal and vertical TE fluxes in the Cape Basin and Argentine Basin along the 40ºS transect: (1) $^{228}$Ra-derived-diffusive; (2) $^{228}$Ra-derived-advective; and (3) TE/$^{228}$Ra-ratio-derived TE fluxes, following the methods in Charette et al. (2016), Sanial et al. (2018) and Vieira et al. (2020); and the shelf $^{228}$Ra fluxes, based on the global seawater $^{228}$Ra inverse models (Kwon et al., 2014; Le Gland et al., 2017). The details of calculations are provided in Appendix D. The results of the TE fluxes are summarized in Table 2 (shelf-ocean, horizontal) and Table 3 (vertical). For comparison, the

horizontal TE fluxes are normalised to the areas of shelf-ocean cross-section (Table D2, Urien and Ewing, 1974; Nelson et al., 1998; Carr and Botha, 2012) (illustrated in Fig. 8) unless otherwise specified.

Surprisingly, the estimates of shelf-ocean TE fluxes show relatively good agreements (within uncertainties) between these three approaches (Table 2), given the limitations of the 1D-$^{228}$Ra-mixing model (Moore, 2015). A similar observation has also been found in the $^{228}$Ra study in the Peruvian continental shelf (Sanial et al., 2018), which suggests that the assumptions

made for the 1D-$^{228}$Ra-mixing model may be not unreasonable. In addition, the TE fluxes show consistent results between the D357 and JC068 data in the Cape Basin. These observations are likely to be a result of the gradients of $^{228}$Ra and TEs





representing a long-term average at an ocean basin scale and being closer to a steady-state condition in the upper water column (e.g. the $^{228}$Ra and TEs in the North Atlantic, Charette et al., 2015).

Comparing the shelf-ocean TE fluxes from this study with previous studies, all the TE fluxes are normalised to the area of
the shelf-ocean cross section. The areas of shelf-ocean cross section from the reference regions are summarised in Table D2 (Emery, 1966; Windom et al., 2006; Hooker et al., 2013; Vieira et al., 2020). In this study, the $^{228}$Ra-derived shelf-ocean Co fluxes range from 3.8 to $22 \times 10^3$ nmol m$^{-2}$ d$^{-1}$ in the Argentine margin and from 4.3 to $6.2 \times 10^3$ nmol m$^{-2}$ d$^{-1}$ in the Cape margin of the 40ºS transect in the South Atlantic. In comparison, previous studies have applied the TE/$^{228}$Ra approach to estimate the shelf Co fluxes in several continental margins: the western North Atlantic ($1.6 \times 10^5$ nmol m$^{-2}$ d$^{-1}$, Charette et
al., 2016), the Peruvian shelf ($1.4 \times 10^5$ nmol m$^{-2}$ d$^{-1}$, Sanial et al., 2018) and the Congo-offshelf 3ºS ($2.8 \times 10^6$ nmol m$^{-2}$ d$^{-1}$, Vieira et al., 2020). Although these fluxes are about one and two orders of magnitude respectively higher than the estimates in the South Atlantic, these regions are also associated with low oxygen which is prone to result in higher Co fluxes. A low shelf-ocean Co flux has been reported in the eastern South Atlantic continental shelf ($11 \sim 18 \times 10^3$ nmol m$^{-2}$ d$^{-1}$, Bown et al., 2011), which is in fact very close to this study region in the Cape Basin.

Along the 40ºS transect, the $^{228}$Ra-derived shelf-ocean Fe fluxes range from 7.9 to $20 \times 10^4$ nmol m$^{-2}$ d$^{-1}$ in the Argentine margin and from 1.2 to $3.1 \times 10^4$ nmol m$^{-2}$ d$^{-1}$ in the Cape margin, which compares well with the estimates of the shelf-ocean Fe flux ($4.5 \times 10^5$ nmol m$^{-2}$ d$^{-1}$) in the western North Atlantic (Charette et al., 2016). However, these fluxes are significantly lower than those values of shelf-ocean Fe fluxes observed in the presence of river plumes (e.g., River Congo; $4.1 \times 10^8$ nmol m$^{-2}$ d$^{-1}$, Vieira et al., 2020), submarine groundwater discharge ($1.3 \times 10^8$ nmol m$^{-2}$ d$^{-1}$, Windom et al., 2006)
and the oxygen minimum zone ($2.1 \times 10^6$ nmol m$^{-2}$ d$^{-1}$, Sanial et al., 2018).

Lastly, the $^{228}$Ra-derived shelf-ocean Zn fluxes range from 2.7 to $6.5 \times 10^4$ nmol m$^{-2}$ d$^{-1}$ in the Argentine and from 0.9 to $1.2 \times 10^4$ nmol m$^{-2}$ d$^{-1}$ in the Cape margins. When compared with the only available shelf-ocean Zn flux value ($1.8 \times 10^6$ nmol m$^{-2}$ d$^{-1}$) in the western North Atlantic (Charette et al., 2016), the Zn fluxes from this study indicate the likely lower boundary of the values in the South Atlantic. This also supports the previous observation that surface water along the 40ºS transect has
some of the lowest reported dissolved Zn concentrations in the global oceans (Wyatt et al., 2014).

From below the mixed layer, the vertical dissolved TE fluxes range from 0.4 to 1.2 nmol Co m$^{-2}$ d$^{-1}$, from 3.6 to 11 nmol Fe m$^{-2}$ d$^{-1}$, and from 13 to 16 nmol Zn m$^{-2}$ d$^{-1}$ along the 40ºS transect (Table 3). These fluxes are consistent with previous estimates of Co in the South Atlantic (0.04 – 0.46 nmol m$^{-2}$ d$^{-1}$, Bown et al., 2011; 0.1 – 4 nmol m$^{-2}$ d$^{-1}$, Rigby et al., 2020) and in the high latitude North Atlantic (0.15 – 0.5 nmol m$^{-2}$ d$^{-1}$, Achterberg et al., 2020), of Fe in the North Atlantic (0.14 –
21.1 nmol m$^{-2}$ d$^{-1}$, Painter et al., 2014), South Atlantic (1 – 27 nmol m$^{-2}$ d$^{-1}$, Rigby et al., 2020) and the Southern Ocean (27 – 135 nmol m$^{-2}$ d$^{-1}$, Dulaiova et al., 2009; 3 – 31 nmol m$^{-2}$ d$^{-1}$, Blain et al., 2007; 2.3 – 14 nmol m$^{-2}$ d$^{-1}$, Charette et al., 2007), and of Zn in the Atlantic (2.7 – 137 nmol m$^{-2}$ d$^{-1}$, Rigby et al., 2020; Achterberg et al., 2020). However, the vertical diffusive Fe fluxes seem generally smaller than the winter mixing Fe fluxes (e.g. 27.3 – 103 nmol m$^{-2}$ d$^{-1}$ in the high latitude North





Atlantic, Achterberg et al., 2018). It is also worth mentioning that the TE fluxes estimated by Blain et al. (2007), Bown et al.
(2011) and Painter et al. (2014) use the $K_z$ values derived from the vertical density profiles instead of $^{228}$Ra.

Atmospheric dust deposition is another important source supplying TEs to the surface ocean. The soluble atmospheric dust
deposition fluxes to the surface 40°S Atlantic are 0.02 – 0.05 nmol Co m$^{-2}$ d$^{-1}$, 1.6 – 5.2 nmol Fe m$^{-2}$ d$^{-1}$ and 0.6 – 6 nmol Zn
m$^{-2}$ d$^{-1}$  as reported by Chance et al. (2015) from the same cruise.

### 4.4 Mass-balance budgets for trace elements in the South Atlantic

The mass-balance budgets of dissolved TEs from different sources (horizontal shelf inputs, vertical upward mixing, and
atmospheric dust deposition) and sinks (exported fluxes) in the surface South Atlantic (40°S transect) are calculated and
summarised in Table 4 and Fig. 8. The vertical upward mixing appears to be a more important source supplying TEs to the
surface water at 40°S compared to atmospheric dust and continental shelf inputs. However, the dominant source or seasonal
variation of the vertical TE inputs cannot be identified in this study. Apart from the internal regeneration, TEs from a
subsurface lateral input from the continental margin can subsequently be brought to the surface by vertical mixing (e.g.
Rijkenberg et al., 2014). Deep winter convective mixing has also been shown as an important source of TEs to the surface
ocean (e.g. Achterberg et al., 2018; 2020; Rigby et al., 2020).

A bulk estimate of the dissolved TE exported fluxes from the surface ocean, supported by new production biological uptake,
can be made using a sinking particulate organic carbon (POC) flux and an estimated TE/C uptake ratio. If we assume that the
$^{234}$Th-derived POC flux is 7 mmol C m$^{-2}$ d$^{-1}$ in this study region (Thomalla et al., 2006),  and that the average cellular TE/C
ratios are 2.2, 63.6, and 7.3  µmol/mol for Co/C, Fe/C and Zn/C respectively in cultured marine phytoplankton (Ho et al.,
2003), the exported fluxes of Co, Fe, and Zn are 15, 445 and 51 nmol m$^{-2}$ d$^{-1}$ ($3.95 \times 10^7$ , $1.15 \times 10^9$ and $1.32 \times 10^8$ mol yr$^{-1}$)
respectively. The exported fluxes are much higher than the net dissolved TE inputs that we have identified in this study
(Table 4). Taking Fe as an example, the total dissolved Fe flux ($1.5 – 4.7 \times 10^7$ mol yr$^{-1}$) only contributes < 4 % of the total
consumption of dissolved Fe, which is far from being enough to balance the iron budget (> 25-fold offset) in the surface
ocean. This could imply that (1) the spatial and temporal variability in $^{234}$Th-derived POC flux is crucial (given that the
integrated time scale is relatively short, the mean-life of $^{234}$Th = 35 days); (2) much lower TE/C ratios are required; or (3)
other sources of TEs need to be considered (e.g. particulate TE or winter deep-mixing).

Previous studies have shown that the $^{234}$Th-derived POC fluxes can vary between 1.5 and 7 mmol C m$^{-2}$ d$^{-1}$ in the South
Atlantic subtropical gyre (Charette and Moran, 1999; Thomalla et al., 2006). The difference is only a factor of 4 to 5, which
may be enough to explain the offsets of Zn (> 2-fold) and Co (> 10-fold) budgets, considering the uncertainties, but it is not
enough to explain the offset of Fe. The range of observed phytoplankton TE/C ratios in the global oceans can vary widely
(e.g. Co/C: 0.00047 – 25.6 µmol/mol; Fe/C: 2.1 – 258 µmol/mol; Zn/C: 0.02 – 110 µmol/mol, Moore et al., 2013). Direct
measurements of the TE/C ratios in the suspended particles in the South Atlantic are required to constrain the removal TE
fluxes. The Fe released from lateral-transport particles has been suggested as a potential source to explain the high dissolved





Fe concentration observed in the upper 800 m waters in the Southwest Atlantic (Rijkenberg et al., 2014). Further studies are needed to understand the TE inputs from lateral-transport particles in the ocean.

**5 Conclusions**

This study investigates the distribution of $^{228}$Ra in the 40°S Atlantic and provides important constraints on ocean mixing and

dissolved TE fluxes (Co, Fe and Zn) to the high productivity region in the South Atlantic. Although the $^{228}$Ra 1-D mixing model shows some limitations in the assumptions, the $^{228}$Ra data do place bounds on maximum mixing rates in this study and the estimates are within the range of observed values in the global oceans. Three different $^{228}$Ra approaches (1D-diffusion, advection and $^{228}$Ra/TE ratio) have been applied to estimate the dissolved TE fluxes to the 40°S Atlantic, and the results are comparable. The net dissolved TE fluxes suggest that vertical upward mixing is more important than atmospheric dust

deposition and continental shelf supply as the main source supplying dissolved TEs to the surface 40°S Atlantic. However, considering the biological uptake, these dissolved TE inputs are far from being enough to balance the TE budgets in the surface ocean of this region, particularly for Fe. Apart from vertical upward mixing, continental shelves and atmospheric dust inputs, other TE inputs (e.g. particulate or winter deep-mixing) may need to be considered to improve our understanding of micronutrient limitations in the high productivity region in the South Atlantic.

**Appendix A. Seawater $^{226}$Ra and silica concentrations**

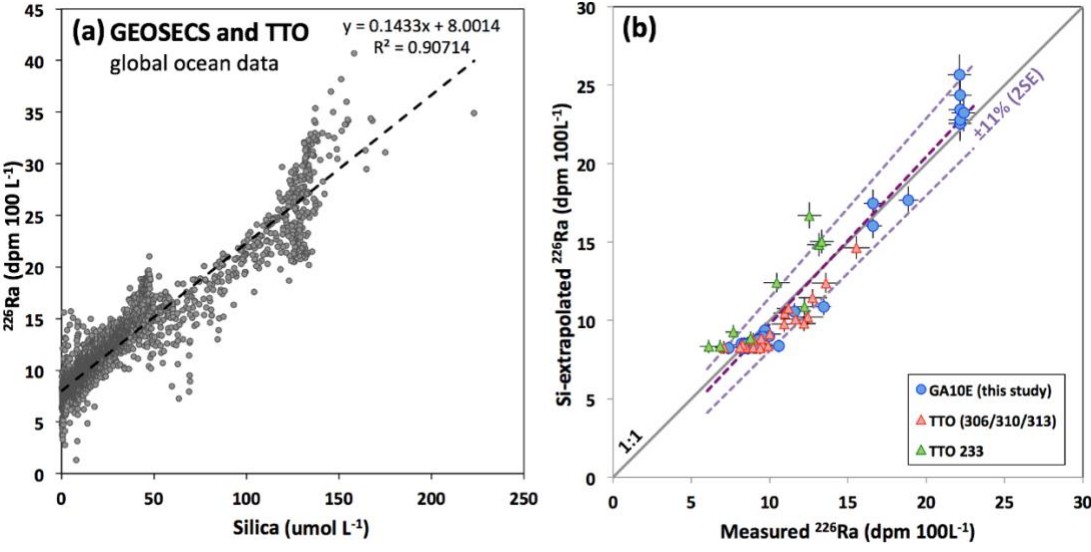

**Figure A1: (a) Relationship between global seawater $^{226}$Ra activity and silica concentration (GEOSECS and TTO datasets). The dashed line shows a linear regression through all the data (b) Plots of Si-extrapolated $^{226}$Ra against measured $^{226}$Ra in this study and the TTO program. The solid line is 1:1, and the purple dashed lines show the linear**
**regressions with the error bars of ±11% (2S.E.)**





## Appendix B. Trace element (TE) data

**Table B1** Surface water (< 10m) dissolved trace element data

| Cruise | Stn | Lon | Lat | dCo (pM) | dFe (nM) | dZn (nM) |
|---|---|---|---|---|---|---|
| D357 | Stn0.5 | 17.606 | -34.336 | | 0.03 | |
| D357 | Stn0.5_r | 17.586 | -34.307 | 30.7 | 0.15 | |
| D357 | Stn1 | 17.054 | -34.620 | 42.2 | | |
| D357 | Stn1_r | 17.010 | -34.617 | 33.4 | | |
| D357 | Stn1.5 | 16.060 | -34.918 | 10.8 | | |
| D357 | Stn2.5 | 14.074 | -35.941 | 7.3 | | |
| D357 | Stn3 | 13.270 | -36.466 | 17.4 | 0.06 | |
| D357 | Stn3.5 | 11.582 | -37.457 | 16.1 | | |
| D357 | Stn4 | 10.052 | -38.416 | 16.5 | | |
| D357 | Stn4.5 | 7.729 | -39.255 | 4.7 | 0.06 | |
| D357 | Stn4.5 | 7.729 | -39.255 | | 0.02 | |
| D357 | FISH_1B | 16.967 | -34.629 | 21.0 | | |
| D357 | FISH_2B | 16.084 | -34.823 | 16.2 | | |
| D357 | FISH_3A | 15.091 | -35.442 | 21.9 | | |
| D357 | FISH_3B | 13.256 | -36.435 | 8.5 | | |
| D357 | FISH_3_Co | 13.129 | -36.499 | 17.0 | | |
| D357 | FISH_4B | 7.688 | -39.302 | 6.5 | | |
| D357 | FISH_4_Co | 10.084 | -38.350 | 17.3 | | |
| D357 | FISH_10B | 17.599 | -34.340 | 19.7 | | |
| D357 | FISH_12_Co | 11.717 | -37.366 | 8.4 | | |
| JC068 | FISH_1001 | 18.128 | -34.044 | 27.7 | 0.30 | 0.14 |
| JC068 | FISH_1002 | 17.921 | -34.154 | 29.3 | 0.30 | 0.09 |
| JC068 | FISH_1003 | 17.726 | -34.237 | 46.9 | 0.35 | 0.07 |
| JC068 | FISH_1004 | 17.452 | -34.392 | 27.3 | 0.13 | 0.06 |
| JC068 | Stn1 | 17.054 | -34.612 | 24.6 | 0.18 | 0.09 |
| JC068 | FISH_1005 | 17.057 | -34.611 | 26.6 | 0.13 | 0.05 |
| JC068 | FISH_1006 | 16.667 | -34.774 | 16.0 | 0.12 | 0.16 |
| JC068 | FISH_1007 | 16.274 | -34.937 | 17.8 | 0.12 | 0.07 |
| JC068 | FISH_1008 | 15.889 | -35.097 | 15.2 | 0.09 | 0.00 |
| JC068 | FISH_1009 | 15.398 | -35.312 | 19.5 | 0.10 | 0.01 |
| JC068 | FISH_1010 | 14.996 | -35.468 | 15.7 | 0.11 | 0.00 |
| JC068 | FISH_1011 | 14.585 | -35.659 | 16.3 | 0.11 | 0.00 |
| JC068 | FISH_1012 | 14.190 | -35.836 | 11.5 | 0.14 | |
| JC068 | FISH_1013 | 13.825 | -35.990 | 17.5 | 0.11 | 0.04 |
| JC068 | FISH_1015 | 13.188 | -36.302 | 18.9 | 0.12 | 0.01 |
| JC068 | Stn3 | 13.104 | -36.348 | 29.7 | 0.18 | |
| JC068 | FISH_1016 | 12.335 | -36.935 | 22.8 | 0.14 | |





| JC068 | FISH_1017 | 11.733 | -37.395 | 14.6 | 0.13 | 0.10 |
|---|---|---|---|---|---|---|
| JC068 | FISH_1018 | 11.086 | -37.890 | 15.2 | 0.16 | 0.01 |
| JC068 | FISH_1019 | 10.443 | -38.368 | 8.1 | 0.16 | 0.03 |
| JC068 | FISH_1020 | 9.778 | -38.603 | 6.0 | 0.13 | 0.04 |
| JC068 | FISH_1021 | 8.606 | -38.992 | 8.3 | 0.15 | 0.04 |
| JC068 | FISH_1022 | 8.378 | -39.069 | 5.1 | 0.20 | |
| JC068 | FISH_1023 | 7.705 | -39.289 | 7.4 | 0.10 | 0.00 |
| JC068 | FISH_1024 | 7.116 | -39.485 | 5.2 | 0.10 | 0.04 |
| JC068 | FISH_1114 | -54.017 | -35.988 | 83.9 | | |
| JC068 | Stn24 | -54.000 | -36.000 | 65.5 | 1.53 | 0.58 |
| JC068 | FISH_1110 | -53.335 | -36.335 | 146.2 | 1.35 | 0.59 |
| JC068 | FISH_1109 | -53.166 | -36.522 | 71.4 | 0.50 | 0.25 |
| JC068 | Stn22 | -53.102 | -36.536 | 36.7 | 1.05 | 0.17 |
| JC068 | FISH_1108 | -53.007 | -36.595 | 111.4 | 0.44 | 0.16 |
| JC068 | FISH_1107 | -52.850 | -36.720 | 59.8 | 0.42 | 0.31 |
| JC068 | FISH_1106 | -52.721 | -36.831 | 47.6 | 0.54 | 0.14 |
| JC068 | FISH_1105 | -52.598 | -36.943 | 10.5 | 0.39 | 0.19 |
| JC068 | Stn21 | -52.497 | -37.019 | 43.2 | 0.22 | 0.03 |
| JC068 | Stn21 | -52.427 | -37.049 | 59.7 | 0.31 | 0.11 |
| JC068 | FISH_1103 | -52.103 | -37.265 | 70.9 | 0.37 | 0.09 |
| JC068 | FISH_1102 | -51.405 | -37.769 | 51.8 | 0.33 | 0.26 |
| JC068 | Stn20 | -50.992 | -38.042 | 19.9 | 0.30 | 0.05 |
| JC068 | FISH_1101 | -51.114 | -38.145 | 16.5 | 0.22 | 0.22 |
| JC068 | FISH_1100 | -50.632 | -38.208 | 18.4 | 0.20 | 0.29 |
| JC068 | FISH_1099 | -49.954 | -38.588 | 10.1 | 0.21 | 0.15 |
| JC068 | FISH_1098 | -49.286 | -38.965 | 17.1 | 0.57 | |
| JC068 | FISH_1097 | -48.464 | -39.422 | 28.2 | 0.62 | 0.24 |
| JC068 | FISH_1096 | -47.540 | -39.923 | 12.8 | 0.66 | 0.26 |
| JC068 | Stn19 | -47.417 | -39.992 | 21.7 | 0.21 | 0.10 |
| JC068 | FISH_1095 | -44.824 | -40.001 | 14.9 | 0.50 | 0.18 |
| JC068 | FISH_1094 | -43.768 | -40.003 | 12.3 | 0.23 | 0.17 |
| JC068 | Stn18 | -42.417 | -40.000 | 16.3 | 0.23 | 0.22 |
| JC068 | Stn18 | -42.297 | -40.000 | 9.2 | 0.23 | 0.25 |

**Table B2** Upper ocean (< 600m) dTE data

| Cruise | Stn | Lat | Lon | Depth (m) | dCo (pM) | dFe (nM) | dZn (nM) |
|---|---|---|---|---|---|---|---|
| D357 | Stn1 | 17.054 | -34.620 | 5 | 41.5 | | |
| D357 | Stn1 | 17.054 | -34.620 | 20 | 40.0 | | |
| D357 | Stn1 | 17.054 | -34.620 | 34 | 38.2 | | |
| D357 | Stn1 | 17.054 | -34.620 | 49 | 48.8 | | |





| | | | | | | |
|---|---|---|---|---|---|---|
| D357 | Stn1 | 17.054 | -34.620 | 49 | 42.6 | |
| D357 | Stn1 | 17.054 | -34.620 | 74 | 54.6 | |
| D357 | Stn1 | 17.054 | -34.620 | 99 | 61.2 | |
| D357 | Stn1 | 17.054 | -34.620 | 148 | 58.8 | |
| D357 | Stn1 | 17.054 | -34.620 | 197 | 48.5 | |
| D357 | Stn1 | 17.054 | -34.620 | 347 | 59.3 | |
| D357 | Stn1 | 17.054 | -34.620 | 495 | 60.8 | |
| D357 | Stn1 | 17.054 | -34.620 | 624 | 54.4 | |
| D357 | Stn2 | 15.000 | -35.467 | 20 | 19.7 | |
| D357 | Stn2 | 15.000 | -35.467 | 35 | 33.0 | |
| D357 | Stn2 | 15.000 | -35.467 | 54 | 31.2 | |
| D357 | Stn2 | 15.000 | -35.467 | 64 | 30.3 | |
| D357 | Stn2 | 15.000 | -35.467 | 84 | 26.7 | |
| D357 | Stn2 | 15.000 | -35.467 | 109 | 40.7 | |
| D357 | Stn2 | 15.000 | -35.467 | 163 | 37.2 | |
| D357 | Stn2 | 15.000 | -35.467 | 361 | 56.5 | |
| D357 | Stn2 | 15.000 | -35.467 | 510 | 41.8 | |
| D357 | Stn3 | 13.117 | -36.333 | 5 | | 0.07 |
| D357 | Stn3 | 13.117 | -36.333 | 10 | 17.7 | |
| D357 | Stn3 | 13.117 | -36.333 | 23 | | 0.06 |
| D357 | Stn3 | 13.117 | -36.333 | 29 | 17.0 | |
| D357 | Stn3 | 13.117 | -36.333 | 47 | | 0.05 |
| D357 | Stn3 | 13.117 | -36.333 | 50 | 17.5 | |
| D357 | Stn3 | 13.117 | -36.333 | 50 | | 0.06 |
| D357 | Stn3 | 13.117 | -36.333 | 69 | 28.0 | |
| D357 | Stn3 | 13.117 | -36.333 | 97 | | 0.09 |
| D357 | Stn3 | 13.117 | -36.333 | 99 | 34.1 | |
| D357 | Stn3 | 13.117 | -36.333 | 119 | 17.2 | |
| D357 | Stn3 | 13.117 | -36.333 | 196 | | 0.14 |
| D357 | Stn3 | 13.117 | -36.333 | 197 | 42.7 | |
| D357 | Stn3 | 13.117 | -36.333 | 218 | 37.1 | |
| D357 | Stn3 | 13.117 | -36.333 | 395 | | 0.30 |
| D357 | Stn3 | 13.117 | -36.333 | 496 | 53.8 | |
| D357 | Stn3 | 13.117 | -36.333 | 594 | | 0.44 |
| D357 | Stn4.5 | 7.800 | -39.217 | 10 | | 0.07 |
| D357 | Stn4.5 | 7.800 | -39.217 | 20 | 1.7 | |
| D357 | Stn4.5 | 7.800 | -39.217 | 23 | | 0.06 |
| D357 | Stn4.5 | 7.800 | -39.217 | 47 | | 0.05 |
| D357 | Stn4.5 | 7.800 | -39.217 | 50 | 7.7 | |
| D357 | Stn4.5 | 7.800 | -39.217 | 74 | 3.5 | |
| D357 | Stn4.5 | 7.800 | -39.217 | 97 | | 0.07 |





| | | | | | | | |
|---|---|---|---|---|---|---|---|
| D357 | Stn4.5 | 7.800 | -39.217 | 99 | 8.7 | 0.06 | |
| D357 | Stn4.5 | 7.800 | -39.217 | 124 | 8.4 | | |
| D357 | Stn4.5 | 7.800 | -39.217 | 149 | 14.1 | | |
| D357 | Stn4.5 | 7.800 | -39.217 | 173 | 16.1 | | |
| D357 | Stn4.5 | 7.800 | -39.217 | 197 | | 0.12 | |
| D357 | Stn4.5 | 7.800 | -39.217 | 198 | 30.1 | | |
| D357 | Stn4.5 | 7.800 | -39.217 | 302 | 28.8 | | |
| D357 | Stn4.5 | 7.800 | -39.217 | 400 | | 0.25 | |
| D357 | Stn4.5 | 7.800 | -39.217 | 500 | 36.9 | | |
| JC068 | Stn1 | 17.054 | -34.612 | 18 | 26.9 | 0.22 | 0.21 |
| JC068 | Stn1 | 17.054 | -34.612 | 29 | 17.5 | 0.14 | 0.13 |
| JC068 | Stn1 | 17.054 | -34.612 | 41 | 22.3 | 0.28 | 0.01 |
| JC068 | Stn1 | 17.054 | -34.612 | 48 | 31.9 | 0.08 | 0.01 |
| JC068 | Stn1 | 17.054 | -34.612 | 58 | 34.9 | 0.16 | 0.02 |
| JC068 | Stn1 | 17.054 | -34.612 | 78 | 32.6 | 0.29 | 0.39 |
| JC068 | Stn1 | 17.054 | -34.612 | 98 | 43.9 | 0.37 | 0.34 |
| JC068 | Stn1 | 17.054 | -34.612 | 197 | 45.4 | 0.35 | 0.34 |
| JC068 | Stn1 | 17.054 | -34.612 | 247 | 44.8 | 0.45 | 0.25 |
| JC068 | Stn1 | 17.054 | -34.612 | 296 | 50.8 | 0.48 | 0.28 |
| JC068 | Stn1 | 17.054 | -34.612 | 398 | 45.2 | 0.54 | 0.49 |
| JC068 | Stn2 | 14.996 | -35.468 | 79 | 22.0 | | 0.15 |
| JC068 | Stn2 | 14.996 | -35.468 | 98 | 32.1 | 0.19 | 0.28 |
| JC068 | Stn2 | 14.996 | -35.468 | 197 | 26.6 | 0.16 | 0.37 |
| JC068 | Stn2 | 14.996 | -35.468 | 246 | 38.0 | 0.20 | 0.38 |
| JC068 | Stn2 | 14.996 | -35.468 | 298 | 39.3 | 0.21 | 0.22 |
| JC068 | Stn2 | 14.996 | -35.468 | 399 | 45.4 | 0.21 | 0.40 |
| JC068 | Stn3 | 13.104 | -36.348 | 19 | 24.9 | 0.13 | 0.21 |
| JC068 | Stn3 | 13.104 | -36.348 | 34 | 36.4 | 0.17 | 0.27 |
| JC068 | Stn3 | 13.104 | -36.348 | 48 | 27.8 | 0.25 | 0.10 |
| JC068 | Stn3 | 13.104 | -36.348 | 74 | 30.7 | 0.14 | 0.21 |
| JC068 | Stn3 | 13.104 | -36.348 | 98 | 36.6 | 0.17 | 0.08 |
| JC068 | Stn3 | 13.104 | -36.348 | 149 | 31.1 | | 0.13 |
| JC068 | Stn3 | 13.104 | -36.348 | 199 | 34.5 | 0.29 | 0.06 |
| JC068 | Stn3 | 13.104 | -36.348 | 298 | 34.2 | 0.33 | 0.20 |
| JC068 | Stn3 | 13.104 | -36.348 | 398 | 38.1 | 0.41 | 0.53 |
| JC068 | Stn3 | 13.104 | -36.348 | 498 | 36.1 | 0.46 | 0.35 |





## Appendix C. Relationship between seawater $^{228}Ra_{ex}$ and salinity in the Argentine Basin

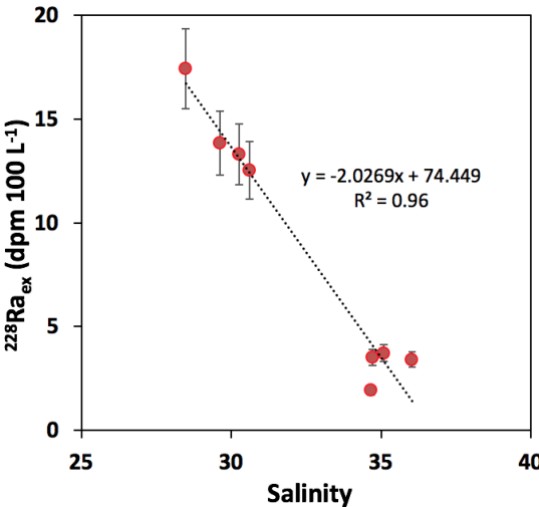


**Figure C1: Relationship between seawater $^{228}Ra_{ex}$ activity and salinity in the Argentine Basin. The dashed line shows the linear regression. The error bars of $^{228}Ra_{ex}$ activity are 2S.E.**

### Appendix D. Trace element (TE) flux calculations

In this study, we use three different $^{228}Ra$ approaches to quantify the horizontal and diffusive vertical dTE fluxes in the Cape

Basin and Argentine

#### (1) $^{228}Ra$-derived-diffusive TE fluxes

To calculate both lateral and vertical TE fluxes, the $^{228}Ra$-derived diffusion coefficients ($K_z$ or $K_x$) are applied to Fick's first

law of molecular diffusion in the following equation:

$$F_{TE-d} = K_{x \, or \, z} \left( {\Delta TE} / {\Delta x \, or \, \Delta z} \right) \tag{D1}$$

where $F_{TE-d}$ is the diffusive flux of the TEs, and $\Delta TE / \Delta x \, or \, \Delta z$ is the gradient of TE concentration over either the horizontal

distance $x$ to the coasts or the vertical depth $z$ below the mixed layer, which can be obtained from the linear regression of

horizontal and vertical TE profiles in Fig. 3 and Fig. 4 respectively. In the Argentine Basin, the cross-shelf horizontal

gradients of TEs (Fig.3a, c and e) and $^{228}Ra$-derived $K_x$ (Fig. 5a) have been used to assess the TE fluxes cross the shelf break

and the Brazil Current. In the Cape Basin, where the TE gradients (Co and Fe) from both cruises D357 and JC068 are used

for comparison (Fig. 3b and d), the differences are generally less than 20% (except for the vertical gradient of Co at Stn3).

As the vertical gradients and $^{228}Ra$-derived $K_z$ in the Argentine Basin are likely to be biased by lateral inputs, the vertical TE

fluxes are not assessed in the western transect here. The $^{228}Ra$-derived-diffusion TE fluxes are summarized in Table 2

(horizontal) and Table 3 (vertical).





*(2) $^{228}$Ra-derived-advective TE fluxes*

As discussed above (in Section 4.1.2), surface water after the boundary of the shelf break and the Brazil Current in the Argentine Basin carries strong offshore advection signals along the SAC towards the open ocean. Assuming that the mixing of TE is conservative, the advective TE fluxes can be calculated using the following equation:

$$F_{TE-a} = w \cdot [TE]_{ave-0} \tag{D2}$$

where $F_{TE-a}$ is the offshore advective flux of TE, $w$ is the net offshore advection velocity ($0.6 \pm 0.3$ cm s$^{-1}$) along the SAC in the Argentine Basin, and $[TE]_{ave-0}$ is the average concentrations of dissolved TEs in the initial advective waters around where the Brazil Current merges into the SAC (around Stn21, Fig. 3): $[Co]_{ave-0} = 40 \pm 21$ pM; $[Fe]_{ave-0} = 0.36 \pm 0.13$ nM; $[Zn]_{ave-0} = 0.12 \pm 0.07$ nM (1S.D., n = 4). The calculated advective TE fluxes are summarised in Table 2 for comparison. As the advective signals cannot be easily separated from the mixing in the Cape Basin, the advective TE fluxes are not assessed in

the eastern transect here.

*(3) TE/$^{228}$Ra-ratio-derived TE fluxes*

Previous studies have combined the use of the shelf $^{228}$Ra fluxes with the ratios of TE/$^{228}$Ra in the surface waters between continental shelves and open oceans to estimate the shelf-ocean TE inputs from the continental margins to the open oceans (Charette et al., 2016; Sanial et al., 2018; Vieira et al., 2020). This method provides the integrated net fluxes of TEs,

considering all the possible inputs (e.g. rivers, SGD and sediments) and outputs (e.g. particle scavenging, biological uptake and radioactive decay) of $^{228}$Ra and TEs during water mixing between the continental shelf and the open ocean. More details of the method are given in Charette et al. 2016. In brief, assuming that the net shelf-ocean exchange is mainly driven by eddy diffusion, the cross-shelf TE fluxes can be calculated using the following equation:

$$F_{TE} = F_{228Ra} \cdot \left(\frac{\Delta TE}{\Delta 228Ra}\right) = F_{228Ra} \cdot \left(\frac{TE_{shelf} - TE_{ocean}}{228Ra_{shelf} - 228Ra_{ocean}}\right) \tag{D3}$$

where $F_{228Ra}$ is the cross-shelf $^{228}$Ra flux; $TE_{shelf}$ and $^{228}$Ra$_{shelf}$ are the average concentrations of the TE and $^{228}$Ra in the surface waters on the shelf (GA10W: between Stn23 and Stn25; GA10E: between Stn0 and Stn1) respectively; and $TE_{oecan}$ and $^{228}$Ra$_{ocean}$ are the average concentrations in the open ocean (GA10W: between Stn18 and Stn19; GA10E: between Stn4 and Stn4.5). The ratios of $\Delta TE/\Delta^{228}$Ra are reported in Table D1.

The shelf $^{228}$Ra flux is based on the inverse models using the global seawater $^{228}$Ra database and inventory (Kwon et al.,

2014; Le Gland et al., 2017). In the South Atlantic, the average shelf $^{228}$Ra flux is $1.7 \pm 0.3 \times 10^{10}$ atoms m$^2$ yr$^{-1}$ around the Uruguayan and South African continental margins (normalised to shelf area; Charette et al., 2016). For comparison, the flux is converted to the shelf-ocean cross sectional flux by multiplying the average continental shelf widths (Cape Basin: 80 km; Argentine Basin: 120 km) and then dividing by the water depths at the shelf break (Cape Basin: 150 m; Argentine Basin: 160





m) (Urien and Ewing, 1974; Nelson et al., 1998; Carr and Botha, 2012). The shelf length is shared between the two surfaces

and cancelled out during the calculation (Fig. 8 and Table D2). The cross-shelf $^{228}$Ra flux ($F_{228Ra}$) becomes $1.3 \pm 0.2 \times 10^{13}$ atoms m$^2$ yr$^{-1}$ in the Argentine Basin and $0.9 \pm 0.2 \times 10^{13}$ atoms m$^2$ yr$^{-1}$ in the Cape Basin. The calculated TE fluxes are summarised in Table 2.

It is important to remember that when considering the sources of $^{228}$Ra and TEs in the ocean, TEs may have their maximum source term at a different depth than $^{228}$Ra. Whereas $^{228}$Ra has a clear maximum from the continental shelf in the surface

mixed layer, redox-sensitive, more particle-bound or hydrothermal-related TEs may see a maximum at deeper levels, due to particle resuspension, low oxygen saturation or hydrothermal activity. In this sense, our model calculations provide the estimates of lateral TE inputs in the surface mixed layer with a clear source from the continental shelf. The vertical TE inputs, however, do not separate the TE inputs between internal cycling, hydrothermal or lateral transport from the continental margin at greater depths. Nevertheless, all these inputs would only become relevant for productivity if they reach

the surface mixed layer later, e.g. by vertical mixing as quantified here.

**Table D1** Shelf and open ocean average dTE and $^{228}$Ra concentrations and $\Delta$TE/$\Delta^{228}$Ra ratios

| | | $^{228}$Ra | dCo | dFe | dZn |
|---|---|---|---|---|---|
| | | ($10^5$ atoms L$^{-1}$) | (pmol L$^{-1}$) | (nmol L$^{-1}$) | (nmol L$^{-1}$) |
| Argentine | Shelf | 6.4 ± 1.1 | 99 ± 42 | 1.44 ± 0.12 | 0.58 ± 0.01 |
| | Open ocean | 1.3 ± 0.5 | 15 ± 5 | 0.28 ± 0.12 | 0.18 ± 0.05 |
| Cape | Shelf | 1.3 ± 0.1 | 30 ± 8 | 0.22 ± 0.09 | 0.08 ± 0.03 |
| | Open ocean | 0.5 ± 0.1 | 9 ± 5 | 0.11 ± 0.06 | 0.05 ± 0.05 |
| $\Delta$TE/$\Delta^{228}$Ra | ($10^{-7}$nmol atom$^{-1}$) | | | | |
| | Argentine margin | | 1.6 ± 1.1 | 22 ± 14 | 7.7 ± 3.9 |
| | Cape margin | | 2.5 ± 1.5 | 12 ± 8 | 3.6 ± 4.2 |

Only JC068 TE data are used. All errors are ±1S.D.




**Table D2** Average shelf width and shelf-break water depth for shelf-ocean dTE flux normalisation

| Location | Shelf width (km) | Water depth (m) | References |
|---|---|---|---|
| South African margin (Cape Basin) | 80 | 150 | Nelson et al., 1998; Carr and Botha, 2012 |
| Uruguayan margin (Argentine Basin) | 120 | 160 | Urien and Ewing, 1974 |
| Western North Atlantic margin | 135 | 132 | Emery, 1966 |
| Peruvian margin | 100 | 200 | Hooker et al., 2013 |
| | Cross section width (km) | Mixed layer depth (m) | |
| Congo River margin* | 300 | 15 | Vieira et al., 2020 |
| Brazilian margin** | 240 | 10 | Windom et al., 2006 |

Shelf-ocean TE or $^{228}$Ra fluxes presented in this study are normalised to the area of shelf-ocean cross section (by default, the cross section at shelf-break = shelf length x shelf-break water depth). To convert the shelf TE or $^{228}$Ra flux (usually normalised by shelf area) from previous studies, the shelf fluxes are multiplied by the shelf width and length, and then divided by the area of cross section. Shelf length should drop off from the flux conversion. *Congo River margin TE fluxes are divided by a defined cross section (the width of river plume x mixed layer depth). **Brazilian margin Fe flux is divided by a defined cross section (the width of a defined coastline x mixed layer depth).


**Data availability.** All the original and supporting data are shown in the manuscript. The data are also available publicly at the GEOTRACES IDP2017 (https://www.bodc.ac.uk/geotraces/data/idp2017/).

**Author contributions.** YTH, WG and GMH designed the radium projects. YTH conducted the Ra-228 and Ra-226

analyses. EMSW conducted the Si measurements. NJW, MCL and EPA contributed the TE data and interpretation. YTH prepared the manuscript with contributions from all co-authors.

**Competing interests.** The authors declare that they have no conflict of interest.

**Acknowledgements.** The authors wish to thank the officers and crew of *RRS Discovery* and *RRS James Cook* for their assistance on the UK-GEOTRACES GA10 cruises (D357 and JC068). We would like to thank Alex Thomas, Andrew

Mason and Steve Wyatt for assistance with mass spectrometry and laboratory support. We would also like to thank Willard Moore, Will Homoky, Yves Plancherel and Christian Schlosser for feedback and discussion at different stages of the manuscript preparation. This study was funded by grants from the UK Natural Environment Research Council for the UK-GEOTRACES GA10 cruises NE/H008497/1 to WG, NE/F017316/1 to GMH, NE/H004475/1 to MCL and NJW, and NE/H004394/1 to EPA.

 

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



**Table 1** $^{226}$Ra and $^{228}$Ra activities, $^{228}$Ra/$^{226}$Ra activity ratios and silica concentration

| Cruise | Stn | Lon | Lat | Depth | Sal | Temp | $^{226}$Ra | | | $^{228}$Ra | | | $^{228}$Ra/$^{226}$Ra | | | Si |
|---|---|---|---|---|---|---|---|---|---|---|---|---|---|---|---|---|
| | | | | (m) | (psu) | (ºC) | (dpm 100 L$^{-1}$) | | | (dpm 100 L$^{-1}$) | | | (activity ratio) | | | (μM) |
| D357 | 0 | 17.967 | -34.183 | 5 | 35.49 | 17.80 | 8.3 | ± | 0.3 | 3.22 | ± | 0.23 | 0.390 | ± | 0.028 | 1.42 |
| D357 | 1 | 17.050 | -34.617 | 5 | 35.49 | 17.95 | 8.3 | ± | 0.3 | 2.94 | ± | 0.19 | 0.354 | ± | 0.023 | 2.58 |
| D357 | 1 | 17.050 | -34.617 | 20 | 35.49 | 17.92 | 8.6 | ± | 0.3 | 2.68 | ± | 0.16 | 0.312 | ± | 0.019 | 2.41 |
| D357 | 1 | 17.050 | -34.617 | 50 | 35.47 | 17.50 | 8.6 | ± | 0.3 | 2.94 | ± | 0.17 | 0.342 | ± | 0.020 | 3.64 |
| D357 | 1 | 17.050 | -34.617 | 100 | 35.29 | 14.27 | 9.4 | ± | 0.4 | 1.19 | ± | 0.09 | 0.127 | ± | 0.010 | 6.02 |
| D357 | 1 | 17.050 | -34.617 | 200 | 35.01 | 11.66 | 9.6 | ± | 0.4 | 0.80 | ± | 0.08 | 0.084 | ± | 0.008 | 6.77 |
| D357 | 1 | 17.050 | -34.617 | 400 | 34.60 | 7.91 | 11.6 | ± | 0.4 | 0.29 | ± | 0.05 | 0.025 | ± | 0.005 | 17.97 |
| D357 | 1 | 17.050 | -34.617 | 1600 | 34.82 | 2.73 | (15.9 | ± | 1.7) | 0.24 | ± | 0.07 | 0.015 | ± | 0.004 | 54.95 |
| D357 | 1 | 17.050 | -34.617 | 2580 | 34.86 | 2.35 | (16.1 | ± | 1.8) | 0.60 | ± | 0.12 | 0.037 | ± | 0.006 | 56.31 |
| D357 | 2 | 15.000 | -35.467 | 5 | 35.58 | 17.94 | 8.3 | ± | 0.3 | 2.71 | ± | 0.18 | 0.327 | ± | 0.021 | 2.75 |
| D357 | 2 | 15.000 | -35.467 | 50 | 35.58 | 17.94 | 8.2 | ± | 0.3 | 2.85 | ± | 0.19 | 0.347 | ± | 0.023 | 3.45 |
| D357 | 2 | 15.000 | -35.467 | 100 | 35.57 | 17.78 | 8.4 | ± | 0.3 | 2.73 | ± | 0.19 | 0.326 | ± | 0.022 | 3.40 |
| D357 | 2 | 15.000 | -35.467 | 400 | 35.25 | 13.31 | 9.3 | ± | 0.3 | 0.76 | ± | 0.08 | 0.082 | ± | 0.008 | 5.31 |
| D357 | 3 | 13.117 | -36.333 | 5 | 35.08 | 13.00 | 9.1 | ± | 0.3 | 1.52 | ± | 0.13 | 0.167 | ± | 0.014 | 2.01 |
| D357 | 3 | 13.117 | -36.333 | 10 | 35.08 | 13.00 | 9.1 | ± | 0.3 | 1.65 | ± | 0.22 | 0.182 | ± | 0.024 | 2.34 |
| D357 | 3 | 13.117 | -36.333 | 20 | 35.08 | 13.00 | 8.8 | ± | 0.3 | | | | | | | 2.46 |
| D357 | 3 | 13.117 | -36.333 | 50 | 35.00 | 12.62 | (8.3 | ± | 0.9) | 2.20 | ± | 0.34 | 0.264 | ± | 0.028 | 2.37 |
| D357 | 3 | 13.117 | -36.333 | 100 | 34.73 | 11.08 | 9.2 | ± | 0.3 | 1.24 | ± | 0.09 | 0.135 | ± | 0.010 | 2.71 |
| D357 | 3 | 13.117 | -36.333 | 200 | 34.65 | 10.26 | 9.3 | ± | 0.3 | 0.61 | ± | 0.05 | 0.065 | ± | 0.005 | 3.20 |
| D357 | 3 | 13.117 | -36.333 | 400 | 34.56 | 8.20 | 9.7 | ± | 0.3 | 0.41 | ± | 0.04 | 0.042 | ± | 0.004 | 9.52 |
| D357 | 3 | 13.117 | -36.333 | 1410 | 34.59 | 3.01 | (17.2 | ± | 1.9) | 0.30 | ± | 0.05 | 0.018 | ± | 0.002 | 64.46 |
| D357 | 3 | 13.117 | -36.333 | 1500 | 34.65 | 2.87 | 16.6 | ± | 0.6 | | | | | | | 66.06 |
| D357 | 3 | 13.117 | -36.333 | 4335 | 34.74 | 1.16 | (22.9 | ± | 2.5) | 0.80 | ± | 0.11 | 0.035 | ± | 0.003 | 104.23 |
| D357 | 3 | 13.117 | -36.333 | 4425 | 34.74 | 1.13 | 22.2 | ± | 0.7 | | | | | | | 105.61 |
| D357 | 3 | 13.117 | -36.333 | 4706 | 34.73 | 1.10 | (23.3 | ± | 2.6) | 0.71 | ± | 0.11 | 0.030 | ± | 0.003 | 106.90 |
| D357 | 3 | 13.117 | -36.333 | 4776 | 34.73 | 1.09 | (23.4 | ± | 2.6) | 2.05 | ± | 0.34 | 0.088 | ± | 0.011 | 107.65 |
| D357 | 3 | 13.117 | -36.333 | 4823 | 34.73 | 1.10 | 22.2 | ± | 0.7 | | | | | | | 123.30 |
| D357 | 3 | 13.117 | -36.333 | 4895 | 34.73 | 1.10 | 22.1 | ± | 0.7 | | | | | | | 125.26 |
| D357 | 4 | 10.400 | -38.400 | 5 | 34.84 | 11.78 | 9.2 | ± | 0.3 | 0.99 | ± | 0.11 | 0.108 | ± | 0.012 | 1.92 |
| D357 | 4 | 10.400 | -38.400 | 700 | 34.25 | 4.59 | 13.4 | ± | 0.5 | 0.26 | ± | 0.03 | 0.019 | ± | 0.003 | 20.07 |
| D357 | 4.5 | 7.800 | -39.217 | 5 | 35.17 | 13.85 | 8.6 | ± | 0.3 | 1.26 | ± | 0.10 | 0.146 | ± | 0.011 | 1.64 |
| D357 | 4.5 | 7.800 | -39.217 | 10 | 35.17 | 13.78 | (8.2 | ± | 0.9) | 0.99 | ± | 0.14 | 0.121 | ± | 0.010 | 1.67 |





| | | | | | | | | | | | | | | | | |
|---|---|---|---|---|---|---|---|---|---|---|---|---|---|---|---|---|
| D357 | 4.5 | 7.800 | -39.217 | 20 | 35.17 | 13.71 | 8.6 | ± | 0.3 | | | | | | | 1.71 |
| D357 | 4.5 | 7.800 | -39.217 | 90 | 35.17 | 13.56 | 7.4 | ± | 0.3 | | | | | | | 1.59 |
| D357 | 4.5 | 7.800 | -39.217 | 200 | 34.97 | 11.87 | 10.6 | ± | 0.4 | 0.91 | ± | 0.07 | 0.086 | ± | 0.007 | 2.57 |
| D357 | 4.5 | 7.800 | -39.217 | 400 | 34.69 | 9.22 | 10.0 | ± | 0.3 | 0.69 | ± | 0.10 | 0.069 | ± | 0.010 | 6.92 |
| D357 | 4.5 | 7.800 | -39.217 | 600 | 34.36 | 6.17 | (9.9 | ± | 1.1) | 0.29 | ± | 0.04 | 0.030 | ± | 0.003 | 13.09 |
| D357 | 4.5 | 7.800 | -39.217 | 2500 | 34.83 | 2.55 | 16.6 | ± | 0.6 | | | | | | | 56.06 |
| D357 | 4.5 | 7.800 | -39.217 | 3241 | 34.84 | 2.22 | (17.1 | ± | 1.9) | 0.47 | ± | 0.08 | 0.027 | ± | 0.004 | 63.43 |
| D357 | 4.5 | 7.800 | -39.217 | 3500 | 34.83 | 2.09 | 18.9 | ± | 0.6 | | | | | | | 67.39 |
| D357 | 4.5 | 7.800 | -39.217 | 4241 | 34.76 | 1.33 | (21.7 | ± | 2.4) | 0.47 | ± | 0.08 | 0.022 | ± | 0.003 | 95.28 |
| D357 | 4.5 | 7.800 | -39.217 | 4500 | 34.74 | 1.16 | 22.2 | ± | 0.8 | | | | | | | 101.64 |
| D357 | 4.5 | 7.800 | -39.217 | 4741 | 34.74 | 1.16 | (22.7 | ± | 2.5) | 0.38 | ± | 0.08 | 0.017 | ± | 0.003 | 102.38 |
| D357 | 4.5 | 7.800 | -39.217 | 5000 | 34.73 | 1.15 | 22.2 | ± | 0.8 | | | | | | | 103.12 |
| D357 | 4.5 | 7.800 | -39.217 | 5141 | 34.73 | 1.16 | (23.4 | ± | 2.6) | 1.07 | ± | 0.16 | 0.046 | ± | 0.005 | 107.45 |
| D357 | 4.5 | 7.800 | -39.217 | 5211 | 34.73 | 1.17 | (23.4 | ± | 2.6) | 1.12 | ± | 0.17 | 0.048 | ± | 0.005 | 107.29 |
| D357 | 4.5 | 7.800 | -39.217 | 5231 | 34.73 | 1.18 | 22.4 | ± | 0.8 | | | | | | | 107.06 |
| JC068 | 18 | -42.416 | -40.001 | 5 | 34.66 | 18.19 | (8.0 | ± | 0.9) | 2.15 | ± | 0.24 | 0.268 | ± | 0.006 | 0.15 |
| JC068 | 18 | -42.416 | -40.001 | 170 | 34.43 | 7.83 | (8.8 | ± | 1.0) | 1.18 | ± | 0.14 | 0.134 | ± | 0.007 | 5.76 |
| JC068 | 18 | -42.416 | -40.001 | 420 | 34.14 | 4.52 | (9.3 | ± | 1.0) | 1.52 | ± | 0.18 | 0.163 | ± | 0.005 | 9.15 |
| JC068 | 19 | -47.417 | -39.994 | 5 | 34.72 | 18.59 | (8.1 | ± | 0.9) | 3.74 | ± | 0.42 | 0.463 | ± | 0.008 | 0.59 |
| JC068 | 20 | -51.029 | -37.983 | 5 | 35.09 | 22.18 | (8.2 | ± | 0.9) | 3.95 | ± | 0.44 | 0.480 | ± | 0.004 | 1.53 |
| JC068 | 21 | -52.503 | -37.026 | 5 | 36.03 | 23.91 | (8.2 | ± | 0.9) | 3.65 | ± | 0.40 | 0.446 | ± | 0.005 | 1.22 |
| JC068 | 21 | -52.503 | -37.026 | 100 | 36.36 | 19.69 | (8.2 | ± | 0.9) | 2.12 | ± | 0.25 | 0.260 | ± | 0.010 | 1.16 |
| JC068 | 21 | -52.503 | -37.026 | 200 | 35.81 | 16.43 | (8.3 | ± | 0.9) | 2.84 | ± | 0.33 | 0.344 | ± | 0.012 | 1.86 |
| JC068 | 21 | -52.503 | -37.026 | 600 | 34.56 | 7.85 | (9.5 | ± | 1.0) | 1.02 | ± | 0.12 | 0.107 | ± | 0.004 | 10.66 |
| JC068 | 22 | -53.102 | -36.538 | 5 | 30.26 | 23.00 | (9.4 | ± | 1.0) | 13.55 | ± | 1.49 | 1.441 | ± | 0.008 | 9.80 |
| JC068 | 23 | -53.337 | -36.338 | 5 | 29.62 | 23.35 | (9.6 | ± | 1.1) | 14.08 | ± | 1.55 | 1.469 | ± | 0.008 | 11.04 |
| JC068 | 24 | -54.000 | -36.000 | 5 | 28.48 | 23.06 | (11.0 | ± | 1.2) | 17.66 | ± | 1.95 | 1.599 | ± | 0.011 | 21.22 |
| JC068 | 25 | -54.560 | -35.493 | 5 | 30.61 | 23.04 | (8.5 | ± | 0.9) | 12.76 | ± | 1.41 | 1.497 | ± | 0.013 | 3.68 |

[226]Ra activity in bracket is extrapolated from the [226]Ra-silica relationship in Fig. A1. [228]Ra activity is calculated from the activity ratio of [228]Ra/[226]Ra multiplied by [226]Ra activity. All errors are 2 standard errors.







**Table 2** $^{228}$Ra-derived shelf-ocean dTE fluxes along the 40°S Atlantic transect

| dTE fluxes | dCo | dFe | dZn |
|---|---|---|---|
| | ($10^3$ nmol m$^{-2}$ d$^{-1}$) | ($10^4$ nmol m$^{-2}$ d$^{-1}$) | ($10^4$ nmol m$^{-2}$ d$^{-1}$) |
| (1) $^{228}$Ra-derived diffusive fluxes | | | |
| Argentine margin (JC068) | 3.8 ± 4.1 | 11 ± 9 | 4.2 ± 3.7 |
| Cape margin (JC068) | 4.7 ± 1.6 | 1.5 ± 0.9 | 1.2 ± 0.7 |
| Cape margin (D357) | 4.3 ± 1.8 | 1.2 ± 1.3 | |
| (2) $^{228}$Ra-derived advective fluxes | | | |
| Argentine margin (JC068) | 22 ± 16 | 20 ± 13 | 6.5 ± 5.0 |
| (3)TE/$^{228}$Ra-ratio-derived fluxes | | | |
| Argentine margin (JC068) | 5.7 ± 4.0 | 7.9 ± 2.7 | 2.7 ± 1.5 |
| Cape margin (JC068) | 6.2 ± 4.0 | 3.1 ± 2.2 | 0.9 ± 1.1 |

Fluxes are normalised to the area of cross-shelf section. All errors are ±1S.D.


**Table 3** $^{228}$Ra-derived vertical dTE fluxes along the 40°S Atlantic transect

| dTE fluxes | dCo | dFe | dZn |
|---|---|---|---|
| Station | (nmol m$^{-2}$ d$^{-1}$) | (nmol m$^{-2}$ d$^{-1}$) | (nmol m$^{-2}$ d$^{-1}$) |
| Stn1 (JC068) | 0.9 ± 0.5 | 11 ± 6 | 16 ± 10 |
| Stn1 (D357) | 0.7 ± 0.5 | | |
| Stn2 (JC068) | 0.9 ± 0.2 | 3.6 ± 3.1 | 16 ± 11 |
| Stn2 (D357) | 1.0 ± 0.3 | | |
| Stn3 (JC068) | 0.4 ± 0.2 | 6.4 ± 2.3 | 13 ± 7 |
| Stn3 (D357) | 0.9 ± 0.4 | 5.9 ± 2.1 | |
| Stn4.5 (D357) | 1.2 ± 0.7 | 6.9 ± 3.9 | |

Fluxes are normalised to the surface area. All errors are ±1S.D.






**Table 4** Net dTE fluxes in the 40°S Atlantic open ocean high productivity zone

| dTE fluxes | dCo | dFe | dZn |
|---|---|---|---|
| | ($10^5$ mol year$^{-1}$) | ($10^6$ mol year$^{-1}$) | ($10^6$ mol year$^{-1}$) |
| (1) Shelf-ocean inputs | | | |
|    Argentine Basin | 0.8 - 4.5 | 1.6 - 4.0 | 0.5 - 1.3 |
|    Cape Basin | 0.9 - 1.3 | 0.3 - 0.6 | 0.2 - 0.3 |
|    Argentine + Cape | 1.7 - 5.8 | 1.9 - 4.6 | 0.7 - 1.6 |
| (2) Vertical mixing | 9.9 - 32 | 9.2 - 27 | 33 - 43 |
| (3) Atmospheric inputs | 0.5 - 1.3 | 4.1 - 13 | 1.6 - 16 |
| (1)+(2)+(3) Total dTE fluxes | **12 - 39** | **15 - 47** | **35 - 60** |
| (4) Removal dTE fluxes | **395** | **1151** | **132** |

The high productivity zone is illustrated in Fig.8. The net dTE fluxes of (1) shelf-ocean inputs (Table 2) are multiplied by the area of cross-shelf section above mixed layer. The net TE fluxes of (2) vertical mixing (Table 3), (3) atmospheric inputs (Chance et al., 2015) and (4) the exported fluxes (assessed from the [234]Th-POC fluxes and TE/C ratios) are multiplied by the surface area.

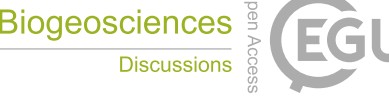


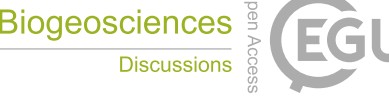

**Figure 1: (a) Map of cruise tracks, station locations and surface currents. GA10E (D357) and GA10W (JC068) stations are labelled with green and red circles respectively. Stations of previous Ra studies are labelled with open symbols (TTO: Key et al., 1990; 1992a,b; GEOSECS: Ku and Lin, 1976; ANT XV/4: Hanfland, 2002). Surface currents are shown with arrows. (b) Salinity profiles along the cruise track of UK-GEOTRACES-GA10, labelled with water masses. Vertical lines indicate the stations where vertical Ra water profiles are available.**





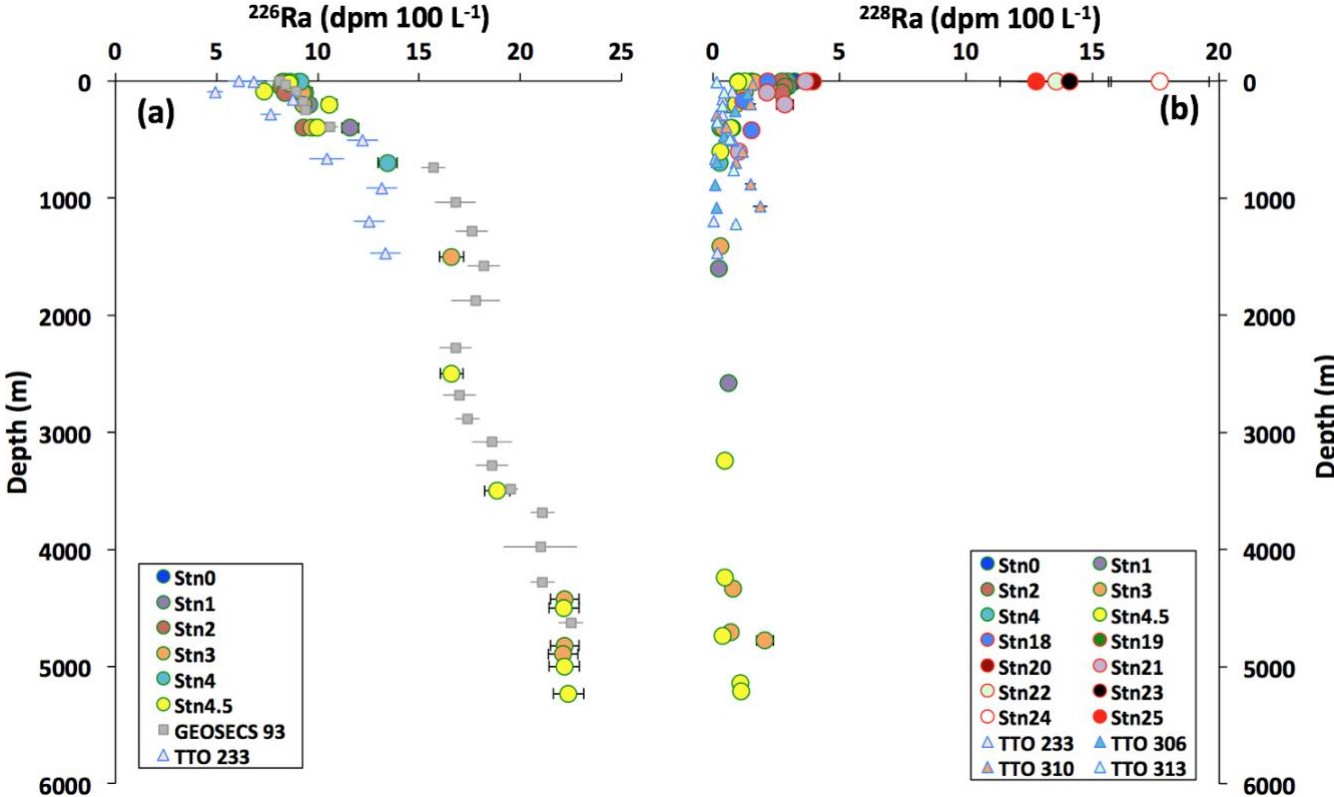


**Figure 2: Depth profiles of (a) $^{226}$Ra and (b) $^{228}$Ra activities. The grey squares show Ra data from the previous GEOSECS study (Ku and Lin, 1976); the triangles show Ra data from the TTO program (Key et al., 1990; 1992a,b). Different water masses are characterised in (a) $^{226}$Ra profile (see details in text). Error bars are ± 2S.E.**





Figure 3: Dissolved trace elements (dCo, dFe and dZn) and salinity in the surface water (< 10 m) along the 40°S Argentine and Cape transects. Red squares show data from cruise JC068 and green circles show data from cruise D357. The orange band indicates the boundary of Brazil Current (BrC) in the Argentine transect, highlighted by high salinity and changing TE gradients. $[TE]_{ave-0}$ is the average concentrations of dissolved TEs in the initial advective waters around where the Brazil Current merges into the SAC (around Stn21; ±1S.D., n = 4). The dashed lines show linear regression trends through the TE data (Argentine transect: only data from the shelf to BrC; Cape transect: the whole transect). The gradient errors are ± 1S.D.







**Figure 4: Depth profiles of dissolved trace elements (dCo, dFe and dZn) in the upper ocean (< 600 m). Red squares show data from cruise JC068 and green circles show data from cruise D357. The dashed lines show linear regression trends with the vertical gradient uncertainty (± 1S.D.)**





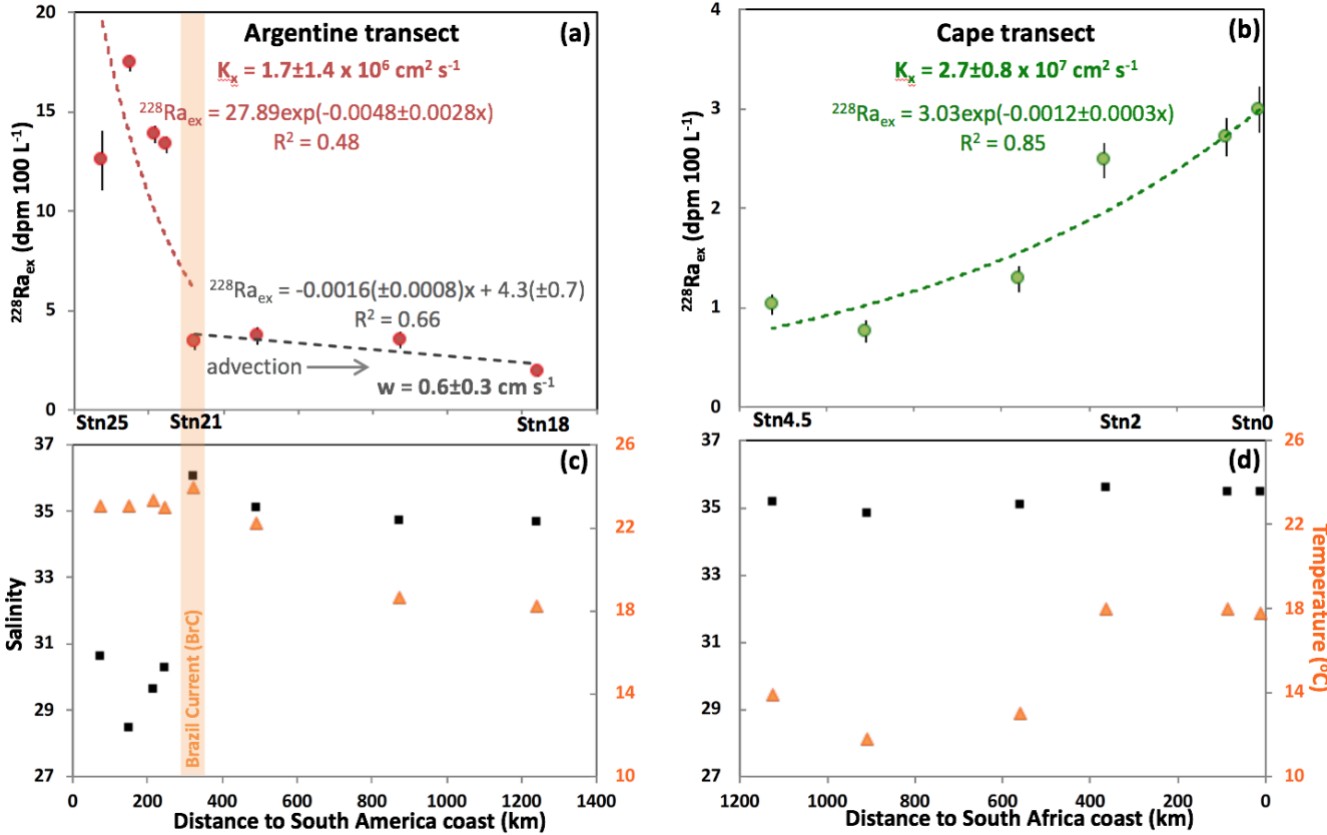


**Figure 5: Plots of $^{228}Ra_{ex}$ in the surface ocean (< 10 m) along the (a) Argentine and (b) Cape transects of 40ºS Atlantic, with the distributions of salinity and temperature shown in (c) and (d). The orange band indicates the boundary of Brazil Current (BrC) in the Argentine transect, highlighted by high salinity and temperature. The red and green dashed lines show exponential regression trends through the $^{228}Ra_{ex}$ data (Argentine transect: only to BrC; Cape transect: the whole transect). The exponential equations**
**are rearranged to match Eqn.3 for the $K_x$ calculation. The errors of $a$ and $K_x$ are ± 1S.D. The grey dashed line shows a linear regression trend through the $^{228}Ra_{ex}$ data from BrC to the open ocean in the Argentine transect, which is used to estimate the advection water transport velocity ($w$).**




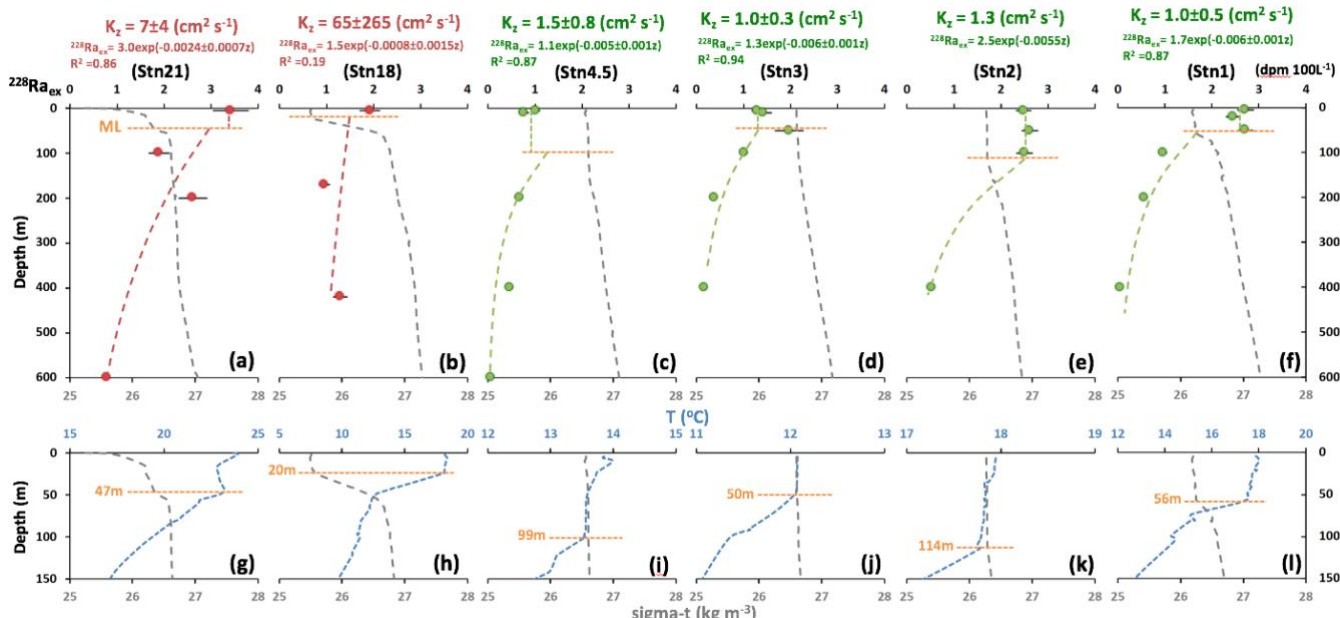

**Figure 6: Depth profiles of (a) to (f) seawater $^{228}Ra_{ex}$ activity in red (Argentine Basin) and green (Cape Basin) circles and density (sigma-t) shown in grey dashed lines, and (g) to (i) density and temperature in the upper ocean shown at Stn1, 2, 3, 4.5, 18 and 21. Depths of mixed layer are labelled with the orange horizontal dashed lines, defined by the sigma-t and temperature profiles. The red and green dashed lines show exponential regression trends through the $^{228}Ra_{ex}$ data below the mixed layer. The exponential equations are used in Eqn.3 for the $K_z$ calculation (errors ± 1S.D.)**

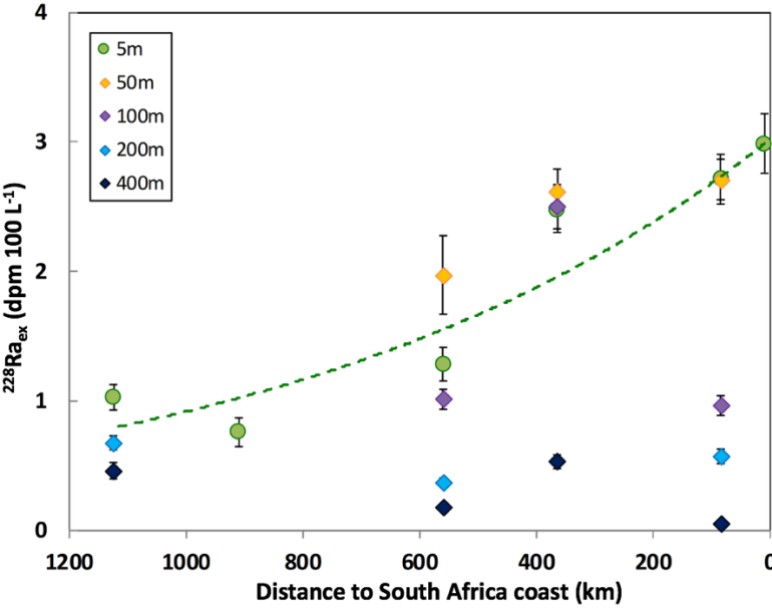

**Figure 7: Plots of $^{228}Ra_{ex}$ activity at water depth 5, 50, 100, 200 and 400 m versus distance to the coast of Cape Town in South Africa. The dashed line shows an exponential regression line through the data at 5 m.**



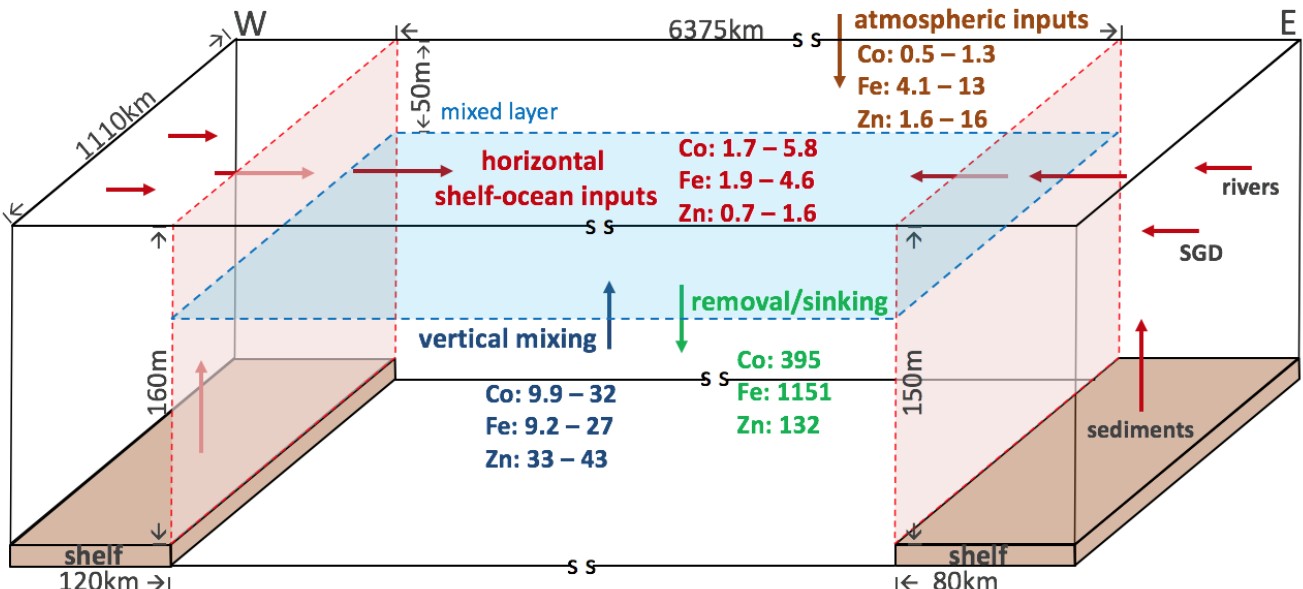

**Figure 8: Schematic diagram of dissolved trace element inputs and outputs in the high productivity zone in the open ocean along the surface 40ºS Atlantic. The approximate dimensions of the high productivity zone and continental shelves are labelled, assuming that the zone spans across the latitude from 35ºS to 45ºS (~ 1110 km) and the longitude from 55ºW to 20ºE (~ 6375 km) with an annual average mixed layer depth of ~ 50 m. The arrows indicate different TE inputs and outputs in this region. The TE fluxes from Table 4 are shown and colour-coded according to the sources, and the units are $10^5$ mol year$^{-1}$ for dCo and $10^6$ mol year$^{-1}$ for dFe and dZn. The vertical red cross sections are used to normalise the shelf-ocean TE and $^{228}$Ra fluxes. The net shelf-ocean TE fluxes represent the total inputs from rivers, SGD and shelf sediments, and the outputs by particle or biological removal between the shelf-ocean mixing from both sides of the continental margins. The blue surface area is used to estimate the net TE inputs from dust and vertical mixing, and the red cross section above the mixed layer is used to estimate the net shelf-ocean TE fluxes (Table 4).**