# Peer review of "Radium-228-derived ocean mixing and trace element inputs in the South Atlantic"

_Biogeosciences, 2020_

## Referee Comment (RC1) · Anonymous Referee #1 · 4 Nov 2020

Hsieh et al (2020) use radium-228 to derive vertical and horizontal mixing rates of trace elements in the South Atlantic. These calculations improve our understanding of trace metal cycling in this part of the ocean, and this manuscript is therefore an important contribution to the field. However, the manuscript could be improved by clarifying when and where certain model assumptions are applied, and by considering some suggested changes/clarifications to the box model calculation. Additionally, I have concerns about the curve fits used to calculate the vertical mixing rates that should be addressed before publication. My comments and suggestions are detailed below, divided by section.

Introduction:

The introduction shifts back and forth from talking about the Southern Atlantic to talking

about nutrient limitation more generally. The authors may wish to re-organize the text so that it starts more broadly and then focuses on the South Atlantic to introduce this specific study. In particular, I recommend moving the second paragraph (lines 41-48) farther down (perhaps making it the second to last paragraph instead), so that there are not two separate "In this study..." statements.

Line 68: the first reference for continental shelf inputs should be "Rutgers van der Loeff et al.", not "van der Loeff et al."

Methods:

It is not clear where each of the cruises started and stopped. Both are described as following a 40 deg S transect, but it is not clear how much overlap there was. It would be helpful to color code the lines on Figure 1a to show each of the cruise tracks (perhaps one color to show JC068 and another color or dashed line to show overlapping sections?).

Line 80: What about the trace metal data? Were all elements measured on both legs?

Line 103: The authors mention that a separate sample is collected for Ra-226 measurements, but do not explain why the larger volume samples cannot be used for this measurement. Please add an explanation of cartridge collection efficiencies and the reason for a separate Ra-226 aliquot.

Three different collection Ra methods are described. Was any intercalibration between methods conducted? (e.g. collecting samples at the same depth using different methods?)

Line 121: The authors explain that the Ra-226 aliquots are spiked with Ra-228. How large is the spike, how can you be sure that no seawater Ra-228 contributes to the "spike" signal? The authors mention that chemical blanks are monitored throughout the procedures, but it is not clear whether this is a seawater sample or a Milli-Q blank. I understand chemical blanks to be reagents only, not including seawater background

activities.

Line 128: The comma after CRM-145 is unnecessary and can be deleted

Line 138: If using a global dataset, why not include more recent GEOTRACES data as well? Alternatively, did the authors consider using an Atlantic-specific trend rather than a global trend? The Ra-226 – Si relationship can vary by basin.

Line 159: Change "on the decade timescales" to "on decadal timescales"

Line 165: The authors state here that vertical mixing could affect horizontal distributions of Ra-228, but that the sample resolution is not good enough to account for this input, and they therefore ignore vertical mixing. This is at odds with the section of the paper where they explicitly use the vertical distribution of Ra-228 to calculate vertical mixing rates. It is not clear to me how they can argue that vertical mixing is insignificant in one case, and the main control on Ra in the second case.

Line 173: This sentence is confusing as written (too many commas) and should be re-phrased.

Line 176: Why is the Ra-228 background determined from the mid-water column? If this is being used to calculate horizontal mixing at the surface, a more appropriate Ra-228 background activity would be the surface water activity in the central South Atlantic (perhaps the ANT XV/4 surface activities? Or GEOTRACES data from the central North Atlantic could also provide a comparison)

Line 185: This sentence states that the two scenarios (mixing only, diffusive only) are used to bracket the range of estimates. However, it is stated in line 193 that the diffusive only case is used nearshore, and the mixing only case is used past the shelf-break. Throughout the methods section, it is confusing which assumptions are applied in which environments (e.g. when vertical mixing is ignored, or when advection is ignored)- perhaps it would help to structure the methods section by geographic region rather than by the individual assumptions?

I am not entirely convinced that an advection-only scenario is valid. In other words, what is the basis for ignoring diffusive mixing? Are there other studies that have used Ra-228 in this way before? To the best of my knowledge, most other Ra-based studies ignore advection rather than diffusion.

Results:

Line 214-215: delete "with expectations based on" so the sentence reads "are consistent with GEOSECS and TTO observations..."

Line 213: Are the authors specifically comparing the vertical profiles with Southern Atlantic GEOSECS and TTO stations here, or the entire Atlantic? If the entire Atlantic, the Charette et al (2015) North Atlantic GEOTRACES data should also be referenced.

Discussion:

Vertical mixing:

Does the exponential curve fit for Ra include any of the data in the surface layer, or does the curve fit start below the mixed layer (Figure 6)?

Station 2 does not have sufficient Ra data (only one point) below the mixed layer to fit a curve. This station should be removed from the vertical mixing analysis.

Some of the linear regressions on the trace element vertical profiles start below the mixed layer, while others do not (Figure 4). Why does this vary from station to station? In particular, the Zn data has particularly poor linear fits, and it is not clear whether all the data were included in the fit (the deepest sample at all three stations are below the end of the dashed line).

Box model/trace element inputs:

Why are the calculations of Ra/TE fluxes put in an appendix? I recommend moving these into the main text, as they form the basis for the conclusions of the paper.

Line 305: change "may not be unreasonable" to "are reasonable"

Lines 309 – 311 are repetitive with line 299-301

Line 319: delete "in fact" so the sentence reads "which is very close.."

Line 321: I recommend saying that the Fe fluxes are "slightly lower than" previously reported fluxes, instead of saying that they "compare well with" these other estimates, as they are an order of magnitude lower.

Line 328: I don't understand why these estimates must represent a lower bound for what is expected in the South Atlantic. If the concentrations of Zn are lower in the Southern Atlantic, wouldn't a lower flux be expected, making these reasonable estimates rather than lower bounds?

Line 335-340: The Southern Ocean Fe fluxes reported by Dulaiova et al (2009) are almost exactly equal to the winter mixing Fe fluxes in the North Atlantic reported by Achterberg et al (2018). However, the authors state that the Fe fluxes they calculated are similar to Dulaiova but lower than Achterberg.

Line 363: stating that the integrated timescale is "relatively short" makes it seem like the half life of Th-234 is too short to capture changes in POC flux. It could be the opposite, that the biology is changing faster than the Th. I suggest removing "the integrated timescale is relatively short" so the sentence reads "given that the mean life of Th is 35 days". This way both possible conditions are accounted for.

The loss via sinking out of the surface box is estimated using 234Th-based POC fluxes, and TE/C uptake ratios in phytoplankton. As I understand it, using the TE/C ratios in phytoplankton aims to isolate the flux from biology. Why not estimate the total flux? The box model sources include inorganic inputs, so the total source from sinking particles should be considered, not just the organic flux. This can be done by multiplying TE/234Th ratios by Th fluxes, rather than adding the extra step of converting to C in the middle. TE/234Th ratios are available for the North Atlantic (Hayes et al., 2018)

and South Pacific (Black et al., 2019). If it is not possible to find TE/234Th ratios for all metals, perhaps this can at least serve as a comparison to the biology-based fluxes for the metals where data is available.

In Hayes et al. (2018), the total Fe flux is significantly smaller than the biological flux estimated here; perhaps the lab-based Fe/C ratios over-estimate the actual ratios in this region?

The depth horizon for the surface box is not mentioned in the text, but is noted as 50 m in Figure 8. Why was this depth chosen? Please add an explanation to the text in addition to noting the depth in the figure. Particle fluxes can vary greatly depending on the depth chosen- see Buesseler et al. 2020 for example.

In the conclusion (line 383) and abstract the authors mention that particle inputs may need to be accounted for to close gaps in the mass balance. What are some examples of these possible particle sources? Is dust deposition not considered a particle source? It seems like some particle inputs and outputs are considered in the box model but not others.

Figures:

Figure 1a: It would be helpful to add "Cape Basin" and "Argentine Basin" to the map, as these locations are referenced throughout the text.

Figure 1a: "GEOSEC" should be changed to "GEOSECS"

Figure 1b: The colorbar extends over a much larger range than necessary; as far as I can tell, there are no samples with a salinity below 34. Shorten the range of the colorbar (e.g. 34 – 36) so that changes in the study area can be more clearly seen. I also recommend using a different color scale- this one has non-linear changes in both hue and brightness that make some gradients appear sharper than others (and will not print well as black & white). The highest peach values can also be misconstrued as orange (falling lower on the color scale between yellow & red, as opposed to being

read as the highest values).

Figure 4: The caption says that dashed lines show linear regression trends with the uncertainty. It's not clear how the uncertainty is shown- should there be shading around the line? Or are the authors referring to the equations next to the lines?

Figure 4: Why do some of the lines not extend through all of the data? (e.g. panels g and j)

Tables:

Table 1: On my PDF, it appears as if half the caption is above the table and half is below. I am not sure if the authors intended to put some information at the end (information about extrapolated Ra-226 activities & errors), but this should be included in the top caption.

Appendix D:

Line 400: The end of this sentence is missing.

Line 444: What two surfaces are the authors referring to?

References:

Buesseler et al. 2020. "Metrics that matter for assessing the ocean biological carbon pump." Proceedings of the National Academy of Sciences. DOI: 10.1073/pnas.1918114117

Hayes et al. 2018. "Flux of Particulate Elements in the North Atlantic Ocean Constrained by Multiple Radionuclides." Global Biogeochemical Cycles. https://doi.org/10.1029/2018GB005994

Black et al. 2019. "Insights From the 238U-234Th Method Into the Coupling of Biological Export and the Cycling of Cadmium, Cobalt, and Manganese in the Southeast Pacific Ocean" Global Biogeochemical Cycles. https://doi.org/10.1029/2018GB005985

---

## Referee Comment (RC2) · Anonymous Referee #2 · 20 Dec 2020

General Comments The scientific quality of this paper is very good and the quantitative results seem both robust and oceanographically consistent. The calculations and argument are concise and the purpose of the paper is very clear. The comparison of different methods used for radium calculations is useful and provides confidence in the results. However, it seems like the radium side of the paper is significantly more developed than the trace metal side. A little more context and detail could be given to the scientific significance of calculating trace element inputs, and more discussion given to the implications of the calculated trace metal fluxes. Also, the methods of the trace metal calculations, which are an integral part of the paper and several sentences of the abstract, could be moved from the appendix to the main body of the paper. One calculation, the Co/C, Fe/C, and Zn/C export fluxes are noted for being significantly

higher than corresponding dissolved inputs. This could be due to the use of laboratory culture metal quota values that were grown under replete conditions. As noted in the paper this region and section has some of the lowest Zn values observed in the euphotic zone from the prior study. Moreover, phytoplankton (and microbes in general) have extraordinary plasticity with respect to metal content, and the lower field and lab scarcity cultures are likely considerably more representative. It would be useful to redo these calculations with metal-carbon values that are more realistic, either using culture studies of phytoplankton grown under scarce rather than replete conditions (e.g. from the relevant Sunda and Huntsman studies) or using particulate metal data from other field expeditions if that data is not available from this section. This would probably help resolve the imbalance in the fluxes compared to the other fluxes calculated and would be more correct in construction of the calculations.

Specific Comments Introduction: Why are we interested in a calculation of Co, Zn, and Fe flux? Why those elements and not others? Please give more context for trace metals in this region, and the significance of the calculations presented here.

Line 40: Define which metals you'll be talking about (Fe, Zn, Co) here, or at least within the first few paragraphs of the introduction, and why they're important in this region.

Lines 83-91: This paragraph should be moved out of the methods section to the introduction or to the results, perhaps as a "hydrographic setting" section in the beginning of the results.

Line 94, 104: Define your trace metal clean technique in more detail. What acids were used and for how long?

Lines 142-5: Useful analysis of the two methods' uncertainty and error differences. I suggest also performing a pairwise t-test to compare the 226Ra data generated from direct observation and from the Si estimation to determine if the two methods are statistically similar, even with the larger error on the estimated method. This would help convince readers that the methods are comparable.

Line 195: Include sections in the methods to clearly describe how Kz was calculated and the vertical flux of trace metals. They are critical results in your paper and are given prominence in your title and abstract, and their methods should not be buried in your appendix. Perhaps an abbreviated description of the appendix D calculations can be described in the methods, and the full version can be in the appendix.

Lines 211-4: Make this point clearer. You imply but don't clearly state that your results are more similar to Atlantic values than Southern Ocean values. That is an interesting result in this transition zone and should be made more explicit.

Lines 215-22: Expand more on this entire section and convey a more detailed picture of TE distributions in this region. Even if the data has been described elsewhere, this section is too short. Additionally, both Fe and Co do not typically continue to "increase with depth below the mixed layer" because they're scavenged-type elements. Fig. 4 implies that all 3 trace elements increase linearly with depth, which is not necessarily the case beyond the mesopelagic. Give more context to the distribution of all three trace metals here, and qualify that the TEs (specifically Fe and Co) only increase with depth in the upper ocean.

Lines 261-7: Move the description of the Kz calculation to the methods and include the explicit equation for its calculation.

Line 298: As already stated, move at least a brief description of the calculation to the methods section.

Line 317: Briefly expand on low oxygen resulting in higher Co fluxes and provide a citation.

Lines 322-5: Discuss your findings more than just comparing them to other fluxes. Why do you think they are lower than the other reported Fe fluxes you cite?

Lines 327-30: Same as the above comment for Fe. Why do you think this region has a lower Zn flux? Was this expected?

Lines 341-3: This paragraph seems out of place. Either expand on why you're reporting aerosol data and give context for the fluxes (how they compare to the vertical TE fluxes, etc.) or simply move these two sentences to section 4.4 where the numbers are used.

Lines 361-72: Specify which of these 3 hypotheses are supported by your results. It sounds like the particles might be a good answer, but it's only mentioned briefly. Expand a bit on what you think is likely going on.

Figure 3 (lines 745-750) The correlations of metals with salinity is very interesting and worth emphasizing a bit more. Similar observations have been observed on the North American shelf as well observed by Bruland and Franks 1984 and Noble et al. 2017. It could be worthwhile to point out that these observations suggest this is a general feature of Atlantic Western boundaries.

Bruland, K. W., & Franks, R. P. (1983). Mn, Ni, Cu, Zn and Cd in the western North Atlantic. In Trace metals in sea water (pp. 395-414). Springer, Boston, MA.

Noble, A. E., Ohnemus, D. C., Hawco, N. J., Lam, P. J., & Saito, M. A. (2017). Coastal sources, sinks and strong organic complexation of dissolved cobalt within the US North Atlantic GEOTRACES transect GA03. Biogeosciences, 14(11).

Technical Comments

General: Keep formatting choices consistent. Both 1-D and 1D are used to describe the model, and spaces are inconsistently placed between a number value and its unit. (ex. line 140 uses 100L-1 and line 141 uses 100 L-1.)

General: Are significant figures to tens of pM for dCo correct?

Line 102: Define RaDeCC here. It is defined later on line 111, and should be defined at the first use of the abbreviation.

Line 114: Define MC-ICP-MS here.

Lines 116-7: Briefly clarify why it is necessary to convert from Sr(Ra)SO4 to

Sr(Ra)CO3.

Line 119: Briefly describe what AG50-X8 and Sr-Spec are. You describe them as ion exchange columns on line 122, but that should be made clearer when they're first mentioned.

Lines 126-7: This sentence was already stated on line 114, but without mention of 226Ra. Are you restating the analyses used, or is this a different method? It is not clear.

Line 127: Change Ra-228 to 228Ra to be consistent with the rest of the paper.

Lines 138-40: This is a run on sentence despite the use of parentheses. I suggest removing the parentheses and forming two separate sentences.

Line 199: "show" not "shows".

Line 253: Use a period or semicolon between "waters" and "therefore" to avoid the run on sentence.

Line 283: Figure 7 only shows depth, not sigma-t as this line states. Rephrase this sentence or relabel the figure to show the density ranges instead of depths.

Line 297: Is the shelf 228fluxes point a fourth part of this list, or is it a separate point from the 3 approaches list? If it's part of the list, label it as (4). If it's a separate point, put it in its own sentence.

Lines 309-10: Confusing sentence – nothing is being compared here. Rephrase.

Figures: The figures here do a good job of supporting the data and conclusion of the work, but they should be edited for readability. In general, do not rely on color coded axes to label subplots (especially for colorblind people), and include axis labels with units clearly next to the axis number scale. Avoid breaking up the axis label and the axis unit. Also, consider putting some of the regression equations in a table or in the appendix, particularly if they're not referenced in the text.

Fig 1: The stations numbers are crammed into the top plot b and are mostly illegible. The Salinity z-axis label is also in a strange location. I suggest leaving more space between plots a and b to include this information above figure b.

Fig 4: Add y-axis labels for depth. Also, the x-axis labels are difficult to find. Is there a way to make the metal label clearer for each row?

Fig 6: The x-axis labels for the top row of plots are not very readable here. The 228Ra label should be next to its units and ideally should be above the axis like the labels for the bottom row of plots. The subtle color coding for sigma-t and T is also confusing.

---

## Author Comment (AC2) · 8 Jan 2021

**Response to Reviewer #2 comments on "Radium-228-derived ocean mixing and trace element inputs in the South Atlantic" by Hsieh et al. (bg-2020-377)**

Reviewer's comments are shown in black.
***Authors' responses are highlighted in blue.

General Comments The scientific quality of this paper is very good and the quantitative results seem both robust and oceanographically consistent. The calculations and argument are concise and the purpose of the paper is very clear. The comparison of different methods used for radium calculations is useful and provides confidence in the results. However, it seems like the radium side of the paper is significantly more developed than the trace metal side. A little more context and detail could be given to the scientific significance of calculating trace element inputs, and more discussion given to the implications of the calculated trace metal fluxes. Also, the methods of the trace metal calculations, which are an integral part of the paper and several sentences of the abstract, could be moved from the appendix to the main body of the paper. One calculation, the Co/C, Fe/C, and Zn/C export fluxes are noted for being significantly higher than corresponding dissolved inputs. This could be due to the use of laboratory culture metal quota values that were grown under replete conditions. As noted in the paper this region and section has some of the lowest Zn values observed in the euphotic zone from the prior study. Moreover, phytoplankton (and microbes in general) have extraordinary plasticity with respect to metal content, and the lower field and lab scarcity cultures are likely considerably more representative. It would be useful to redo these calculations with metal-carbon values that are more realistic, either using culture studies of phytoplankton grown under scarce rather than replete conditions (e.g. from the relevant Sunda and Huntsman studies) or using particulate metal data from other field expeditions if that data is not available from this section. This would probably help resolve the imbalance in the fluxes compared to the other fluxes calculated and would be more correct in construction of the calculations.

***We thank the reviewer for their positive feedback and constructive comments, particularly for the discussion of trace elements. We respond to all the comments point by point below and explain how we will address the issues in the revised manuscript. Following the reviewer's suggestion, we will also apply a range of reasonable estimates for the TE/C ratios to recalculate the TE removal fluxes in this region.

Specific Comments Introduction: Why are we interested in a calculation of Co, Zn, and Fe flux? Why those elements and not others? Please give more context for trace metals in this region, and the significance of the calculations presented here.

***Iron, zinc and cobalt are known to be essential micronutrients for marine phytoplankton. They all play important roles in the cellular metabolic enzymes and sometimes can collimate primary productivity in different ocean regions. Previous studies have shown that the study region near the 40oS Atlantic is mainly iron limited (Moore et al., 2004; Browning et al., 2014; 2017) and has the lowest reported dissolved zinc concentrations in the global ocean (Wyatt et al., 2014). The requirement for Zn in phytoplankton can be replaced by Co in low Zn areas (Price and Morel, 1990). Thus, knowing the sources and fluxes of these three elements can improve our understanding of the limiting factors for productivity in this study region, one of

the most dynamic nutrient regimes in the oceans. We will follow the reviewer's suggestion and add more context in the introduction to highlight the importance of these three elements in this region.

Line 40: Define which metals you'll be talking about (Fe, Zn, Co) here, or at least within the first few paragraphs of the introduction, and why they're important in this region.

***As mentioned above, we will clarify this and add more context in the introduction to highlight the importance of these elements in this region.

Lines 83-91: This paragraph should be moved out of the methods section to the intro- duction or to the results, perhaps as a "hydrographic setting" section in the beginning of the results.

***We agree with the reviewer, but we feel that this paragraph does not fit in either the introduction or the results section. To address this issue, we will thus change the methods section title to "2 Study sites and methods". We will then split the relevant subsection into two, "2.1 Hydrographic setting" and "2.2 Water sampling".

Line 94, 104: Define your trace metal clean technique in more detail. What acids were used and for how long?

***All the trace-metal cleaning procedures follow the GEOTRACES sampling protocols (Cutter et al., 2010). The sample tubing and bottles were rinsed with Milli-Q water and filled with 0.1M HCl for one day. After emptying the acid, the tubing and bottles were rinsed thoroughly with Milli-Q water. The tubing and bottles were also rinsed with open-ocean seawater before sampling. We will add more details in the revised manuscript.

Lines 142-5: Useful analysis of the two methods' uncertainty and error differences. I suggest also performing a pairwise t-test to compare the 226Ra data generated from direct observation and from the Si estimation to determine if the two methods are statistically similar, even with the larger error on the estimated method. This would help convince readers that the methods are comparable.

***Follow the reviewer's suggestion, we have performed a paired t-test and the p-value is 0.55 (> 0.05), which confirms that the measured Ra-226 and the Si-extrapolated Ra-226 data are not statistically significantly different. We will add the t-test results in the revised manuscript.

Line 195: Include sections in the methods to clearly describe how Kz was calculated and the vertical flux of trace metals. They are critical results in your paper and are given prominence in your title and abstract, and their methods should not be buried in your appendix. Perhaps an abbreviated description of the appendix D calculations can be described in the methods, and the full version can be in the appendix.

***We will add more details to describe the calculation of Kz in Section 2.4. As discussed in the response to Reviewer 1 comments, we will summarise the basics of the flux calculations

in the main text by adding a new section in the methods (2.5) and keep the calculation details in the appendix.

Lines 211-4: Make this point clearer. You imply but don't clearly state that your results are more similar to Atlantic values than Southern Ocean values. That is an interesting result in this transition zone and should be made more explicit.

***We will rewrite these sentences to clarify this.

Lines 215-22: Expand more on this entire section and convey a more detailed picture of TE distributions in this region. Even if the data has been described elsewhere, this section is too short. Additionally, both Fe and Co do not typically continue to "increase with depth below the mixed layer" because they're scavenged-type elements. Fig. 4 implies that all 3 trace elements increase linearly with depth, which is not necessarily the case beyond the mesopelagic. Give more context to the distribution of all three trace metals here, and qualify that the TEs (specifically Fe and Co) only increase with depth in the upper ocean.

***We will add more context to describe the distribution of these three trace elements in the study region. We will also clarify that the increase in TE concentrations with depth only applies to the upper ocean.

Lines 261-7: Move the description of the Kz calculation to the methods and include the explicit equation for its calculation.

***We will move the Kz calculation to the method section (2.4) and add more details to describe the calculation.

Line 298: As already stated, move at least a brief description of the calculation to the methods section.

***As mentioned above, we will summarise the basics of the flux calculations in the method section (2.5).

Line 317: Briefly expand on low oxygen resulting in higher Co fluxes and provide a citation.

***We will add a few sentences to explain the relationship between oxygen and cobalt fluxes with relevant references.

Lines 322-5: Discuss your findings more than just comparing them to other fluxes. Why do you think they are lower than the other reported Fe fluxes you cite?

***As already mentioned in the manuscript, these high-Fe fluxes are particularly found in regions with river plumes, SGD and the oxygen minimum zone. We will clarify this in the revised manuscript.

Lines 327-30: Same as the above comment for Fe. Why do you think this region has a lower Zn flux? Was this expected?

\*\*\*There is very limited oceanic Zn flux available for comparison. The only available estimate from the western North Atlantic (Charette et al., 2016) indicates that the cross-shelf Zn fluxes in the South Atlantic are lower than the western North Atlantic. The results may explain why the South Atlantic has the lowest reported Zn concentration in the global oceans. However, it still needs more detailed studies to understand the reasoning behind the different Zn fluxes, which is beyond the scope of this study. We will highlight this in the revised manuscript.

Lines 341-3: This paragraph seems out of place. Either expand on why you're reporting aerosol data and give context for the fluxes (how they compare to the vertical TE fluxes, etc.) or simply move these two sentences to section 4.4 where the numbers are used.

\*\*\*We will move this paragraph to the beginning of the section, and emphasise that apart from the atmospheric dust deposition, other inputs are still poorly constrained in this region.

Lines 361-72: Specify which of these 3 hypotheses are supported by your results. It sounds like the particles might be a good answer, but it's only mentioned briefly. Expand a bit on what you think is likely going on.

\*\*\*As discussed in the response to Reviewer 1 comments, the particle hypothesis is based on the observation from Rijkenberg et al. (2014). They have observed high dissolved Fe in the mid depths of this region and suggested that the laterally transported particles from the offshore export waters may release Fe to the upper ocean. However, it is not possible to provide more discussion without further details of the sources and TE concentrations of these particles and the mechanism releasing TEs from these particles. We will highlight this in the revised manuscript.

Figure 3 (lines 745-750) The correlations of metals with salinity is very interesting and worth emphasizing a bit more. Similar observations have been observed on the North American shelf as well observed by Bruland and Franks 1984 and Noble et al. 2017. It could be worthwhile to point out that these observations suggest this is a general feature of Atlantic Western boundaries.

\*\*\*These observations imply that the distribution of trace elements in the surface ocean is strongly associated with the shelf-water masses in the western boundaries of the Atlantic. We will add a few sentences to highlight this in the revised manuscript.

Bruland, K. W., & Franks, R. P. (1983). Mn, Ni, Cu, Zn and Cd in the western North Atlantic. In Trace metals in sea water (pp. 395-414). Springer, Boston, MA.
Noble, A. E., Ohnemus, D. C., Hawco, N. J., Lam, P. J., & Saito, M. A. (2017). Coastal sources, sinks and strong organic complexation of dissolved cobalt within the US North Atlantic GEOTRACES transect GA03. Biogeosciences, 14(11).

Technical Comments
General: Keep formatting choices consistent. Both 1-D and 1D are used to describe the model, and spaces are inconsistently placed between a number value and its unit. (ex. line 140 uses 100L-1 and line 141 uses 100 L-1.)

***We will check the formatting and ensure consistency in the revised manuscript.

General: Are significant figures to tens of pM for dCo correct?

***This is correct.

Line 102: Define RaDeCC here. It is defined later on line 111, and should be defined at the first use of the abbreviation.

***We will correct this in the revised manuscript.

Line 114: Define MC-ICP-MS here.

***We will define this in the revised manuscript.

Lines 116-7: Briefly clarify why it is necessary to convert from Sr(Ra)SO4 to Sr(Ra)CO3.

***As sulfate is much harder to redissolve back in solutions, the conversion from sulfate to carbonate can improve the efficiency of dissolution. We will clarify this in the revised manuscript.

Line 119: Briefly describe what AG50-X8 and Sr-Spec are. You describe them as ion exchange columns on line 122, but that should be made clearer when they're first mentioned.

***We will clarify these in the revised manuscript.

Lines 126-7: This sentence was already stated on line 114, but without mention of 226Ra. Are you restating the analyses used, or is this a different method? It is not clear.

***This sentence is redundant. We will rewrite and combine it with the next sentence in the revised manuscript.

Line 127: Change Ra-228 to 228Ra to be consistent with the rest of the paper.

***As mentioned above, this sentence will be rewritten and therefore it will not be a problem. However, if an isotope is used in the beginning of a sentence, conventionally the element goes first to avoid beginning the sentence with a number.

Lines 138-40: This is a run on sentence despite the use of parentheses. I suggest removing the parentheses and forming two separate sentences.

***We will rewrite the sentence.

Line 199: "show" not "shows".

***We will correct this.

Line 253: Use a period or semicolon between "waters" and "therefore" to avoid the run on sentence.

***We will make the change.

Line 283: Figure 7 only shows depth, not sigma-t as this line states. Rephrase this sentence or relabel the figure to show the density ranges instead of depths.

***We will rephrase the sentence.

Line 297: Is the shelf 228fluxes point a fourth part of this list, or is it a separate point from the 3 approaches list? If it's part of the list, label it as (4). If it's a separate point, put it in its own sentence.

***The shelf 228Ra flux is only used for the TE/228Ra-ratio-derived TE fluxes. The sentence will be rewritten to clarify this in the revised manuscript.

Lines 309-10: Confusing sentence – nothing is being compared here. Rephrase.

***The sentence will be rephrased to avoid confusion.

Figures: The figures here do a good job of supporting the data and conclusion of the work, but they should be edited for readability. In general, do not rely on color coded axes to label subplots (especially for colorblind people), and include axis labels with units clearly next to the axis number scale. Avoid breaking up the axis label and the axis unit. Also, consider putting some of the regression equations in a table or in the appendix, particularly if they're not referenced in the text.

***We will check and correct these problems in the revised manuscript.

Fig 1: The stations numbers are crammed into the top plot b and are mostly illegible. The Salinity z-axis label is also in a strange location. I suggest leaving more space between plots a and b to include this information above figure b.

***We will make the changes.

Fig 4: Add y-axis labels for depth. Also, the x-axis labels are difficult to find. Is there a way to make the metal label clearer for each row?

***We will add the depth label and clarify the metal label for each row.

Fig 6: The x-axis labels for the top row of plots are not very readable here. The 228Ra label should be next to its units and ideally should be above the axis like the labels for the bottom row of plots. The subtle color coding for sigma-t and T is also confusing.

***We will remake the figure to avoid these problems.

***Other references cited in the response:

Browning et al. (2014) Nutrient regimes control phytoplankton ecophysiology in the South Atlantic, Biogeosciences, 11, 463-479.

Browning et al. (2017) Iron limitation of microbial phosphorus acquisition in the tropical North Atlantic, Nature Communications, 8, 15465.

Charette et al. (2016) Coastal ocean and shelf-sea biogeochemical cycling of trace elements and isotopes: lessons learned from GEOTRACES, Phil. Trans. R. Soc. A, 374, 20160076.

Cutter et al. (2010) Sampling and sample-handling protocols for GEOTRACES cruises. http://www.geotraces.org/libraries/documents/Intercalibration/Cookbook.pdf

Moore et al. (2004) Upper ocean ecosystem dynamics and iron cycling in a global three-dimensional model. Global Biogeochemical Cycles, 18, GB4028.

Price and More (1990) Cadmium and cobalt substitution for zinc in a marine diatom, Nature, 344, 658-660.

Rijkenberg et al. (2014) The distribution of dissolved iron in the west Atlantic Ocean, PLoS One, 9, e101323.

Wyatt et al. (2014) Biogeochemical cycling of dissolved zinc along the GEOTRACES South Atlantic transect GA10 at 40°S, Global Biogeochem. Cy., 28, 44-56.

---

## Author Response (AR1)

**Point-by-point response to Reviewers' comments on "Radium-228-derived ocean mixing and trace element inputs in the South Atlantic" by Hsieh et al. (bg-2020-377)**

Dear Editor,

Thank you for giving us the opportunity to submit the revision of this manuscript. We have uploaded the files of the revised manuscript. A point-by-point response to the reviewers' comments is shown below. Following the suggestions from reviewers, we have rewritten the introduction (section 1.1). In the method section, we have clarified the assumptions and models with more details. We have also moved some information of the trace element (TE) flux calculations from the appendix into the main text (section 2.5). Most importantly, we have fixed the issue of the TE data curve fits and applied a reasonable range of TE/C ratios to recalculate the TE removal fluxes. We believe that these changes have addressed the concerns raised by reviewers, and have clarified any areas of confusion in the previous manuscript. We hope that the revised manuscript is acceptable for publication in *Biogeosciences*.

Reviewer's comments are shown in black.
***Authors' responses are highlighted in blue, and the line numbers refer to the clean version of the revised manuscript (without mark-up).

**Reviewer #1**

Hsieh et al (2020) use radium-228 to derive vertical and horizontal mixing rates of trace elements in the South Atlantic. These calculations improve our understanding of trace metal cycling in this part of the ocean, and this manuscript is therefore an important contribution to the field. However, the manuscript could be improved by clarifying when and where certain model assumptions are applied, and by considering some suggested changes/clarifications to the box model calculation. Additionally, I have concerns about the curve fits used to calculate the vertical mixing rates that should be addressed before publication. My comments and suggestions are detailed below, divided by section.

***We thank the reviewer for their positive feedback and constructive comments. We respond to all the comments point by point below and explain how we have addressed the issues in the revised manuscript. In particular, we have addressed the issue of the curve fit when using the linear regression in Excel. This fit was performed mistakenly assuming that (by default) the TE concentration (x-axis) is the dependent variable and that the depth (y-axis) is the independent variable, but it should be the other way round. This issue has now been corrected in the revised Fig. 4. Most of the gradients stay the same (except for Zn, slightly steeper), and hence the changes in the TE fluxes are insignificant. We have updated the figures and results in the revised manuscript.

Introduction:
The introduction shifts back and forth from talking about the Southern Atlantic to talking about nutrient limitation more generally. The authors may wish to re-organize the text so that it starts more broadly and then focuses on the South Atlantic to introduce this specific study. In particular, I recommend moving the second paragraph (lines 41-48) farther down (perhaps

making it the second to last paragraph instead), so that there are not two separate "In this study. . ." statements.

***We have followed the reviewer's suggestion and rewritten the introduction (Line 32-43; 65-72).

Line 68: the first reference for continental shelf inputs should be "Rutgers van der Loeff et al.", not "van der Loeff et al."

***We have made the correction in the revised manuscript (Line 63).

Methods:
It is not clear where each of the cruises started and stopped. Both are described as following a 40 deg S transect, but it is not clear how much overlap there was. It would be helpful to color code the lines on Figure 1a to show each of the cruise tracks (perhaps one color to show JC068 and another color or dashed line to show overlapping sections?).

***In the revised Fig. 1a, we have added different colour lines (Red: JC068; Green: D357) to show the cruise tracks.

Line 80: What about the trace metal data? Were all elements measured on both legs?

***All the trace elements have been measured on both legs, and most of the trace element data have been published (Wyatt et al., 2014 and 2020; Browning et al., 2014; Clough et al., 2016). We have added more details to clarify that in the revised manuscript (Line 94-95; 111-116; 290-302).

Line 103: The authors mention that a separate sample is collected for Ra-226 measurements, but do not explain why the larger volume samples cannot be used for this measurement. Please add an explanation of cartridge collection efficiencies and the reason for a separate Ra-226 aliquot.

***As the reviewer is already aware of the collection efficiency issues when using the Mn-fibre cartridges, the direct use of large volume samples requires the recovery correction to obtain accurate concentrations of each Ra isotope. The cartridge efficiency can vary hugely, depending on the conditions of fibre coating and the pump flow rates. For example, the efficiency ranges from 70 to 128% in some of the samples in this study (comparing the RaDeCC and MC-ICP-MS Ra-226 results, Geibert et al., 2013).
      Moreover, depending on the instrumentation, other recovery issues may also occur during the sample preparations. For example, MC-ICP-MS requires the purification of Ra from sample matrices prior to the analysis (e.g. fibre leaching and ion chromatography), but these processes could introduce an additional loss of Ra (i.e. 70-90% yield, Hsieh and Henderson, 2011) and hence contribute a large uncertainty in Ra concentrations when using the Ra counts directly from these large volume samples.
      Isotope ratios (or isotope dilution) provide the advantage of being able to correct for the recovery issues, making the valid assumption that recovery does not change the isotope ratios. Thus, the isotope dilution (Ra-228 spike) was used in separate Ra-226 samples to get

accurate Ra-226 concentrations in this study. For the large volume samples, we then only need to focus on measuring Ra-228/226 ratios without being concerned with the recovery issues. We have added more information in the revised manuscript to clarify the reason for and advantage of using a separate Ra-226 aliquot (Line 105-110).

Three different collection Ra methods are described. Was any intercalibration between methods conducted? (e.g. collecting samples at the same depth using different methods?)

***There is no direct intercalibration between the three sampling methods in this study. However, the samples collected by pump (fish), CTD and SAP within the mixed layer at each station show consistent Ra results. We have added a few sentences to clarify this in the revised text (Line 286-288).

Line 121: The authors explain that the Ra-226 aliquots are spiked with Ra-228. How large is the spike, how can you be sure that no seawater Ra-228 contributes to the "spike" signal?

***We usually added ~70 attomol Ra-228 spike to ~250mL seawater. The contribution of seawater Ra-228 is < 0.05 attomol (< 0.1% of the spike signal). We have added more details in the revised manuscript (Line 136-137).

The authors mention that chemical blanks are monitored throughout the procedures, but it is not clear whether this is a seawater sample or a Milli-Q blank. I understand chemical blanks to be reagents only, not including seawater background activities.

***The procedure blanks were carried out in the same way as the samples except there was no added seawater. It is similar to a Milli-Q blank as the reviewer has described, but it also includes all the reagents and procedures that were involved in the sample preparation. We have added some information to clarify this in the revised manuscript (Line 137-140).

Line 128: The comma after CRM-145 is unnecessary and can be deleted

***We have made the correction (Line 142).

Line 138: If using a global dataset, why not include more recent GEOTRACES data as well? Alternatively, did the authors consider using an Atlantic-specific trend rather than a global trend? The Ra-226 – Si relationship can vary by basin.

***Follow the reviewer's comments, we have included more recent GEOTRACES data from the North Atlantic (GA03) and removed the non-Atlantic data. As the non-Atlantic data were only 126 out of 3392 data points (< 4%), the trend slope and interception remain similar to our previous estimates. The difference in the corrected Ra-226 is less than 3%. We have updated the data in the revised manuscript (Line 151-161; 473-479).

Line 159: Change "on the decade timescales" to "on decadal timescales"

***We have made the change (Line 175).

Line 165: The authors state here that vertical mixing could affect horizontal distributions of Ra-228, but that the sample resolution is not good enough to account for this input, and they therefore ignore vertical mixing. This is at odds with the section of the paper where they explicitly use the vertical distribution of Ra-228 to calculate vertical mixing rates. It is not clear to me how they can argue that vertical mixing is insignificant in one case, and the main control on Ra in the second case.

***These model assumptions are justified under different conditions, depending on the mixing time scales and the sources of Ra-228 at the defined sections of the ocean. For example, vertical mixing is typically 5 to 8 orders of magnitude smaller than horizontal mixing. Therefore, the vertical term is often ignored in the Ra-228 horizontal mixing model. In the Ra-228 vertical mixing model, surface waters need to be assumed as the dominant source of Ra-228 to the waters below the mixed layer with no significant horizontal inputs at depths. Hence, the Ra-228 depth gradient is mainly driven by vertical mixing. In the upper ocean, horizontal Ra-228 largely comes from the continental margins (shelf and slope sediments). There is also no direct evidence suggesting that the Ra-228 profiles are affected by the horizontal input below the mixed layer (Fig. 7). Therefore it is not unreasonable to assume that, at least in the top 600 m depth, the vertical gradient of Ra-228 is mainly set by the surface Ra-228 values and downward mixing.

To a certain degree, the Ra-228 background correction in the vertical profiles has considered the horizontal Ra-228 signal at depths. However, this does not mean that all the additional Ra-228 inputs at depths can be corrected. For example, the elevated Ra-228 signals can still be seen at depths in the Argentine Basin and this has led to unreasonably high vertical mixing coefficients. Thus, we do not use the vertical mixing results in these stations.

We acknowledge the fact that these mixing results may only provide upper or lower bound estimates because of the restrictions in these assumptions. Nevertheless, the consistent results shown between this study and previous studies suggest that these assessments are still valuable and can improve our understanding of the trace element budgets in the South Atlantic. We have clarified these assumptions and explain their limitations in the revised manuscript (Line 179-194).

Line 173: This sentence is confusing as written (too many commas) and should be re-phrased.

***We have rewritten the sentence (Line 198-200).

Line 176: Why is the Ra-228 background determined from the mid-water column? If this is being used to calculate horizontal mixing at the surface, a more appropriate Ra-228 background activity would be the surface water activity in the central South Atlantic (perhaps the ANT XV/4 surface activities? Or GEOTRACES data from the central North Atlantic could also provide a comparison)

***In this study, the Ra-228 background is determined by the average value from both the mid-water column and the surface waters (ANT XV/4) far away from shores, because the background correction is applied to both horizontal and vertical mixing. In fact, the Ra-228 background in the mid-water column (1000-3000 m) is not zero (~0.2 dpm/100L) and similar to the remote surface waters (~0.2 dpm/100L), suggesting that there may be an advective Ra-

228 background in the mid-water column and that this should be corrected for in the vertical mixing calculation.

For comparison, we select the surface water data (< 100 m) from the GEOTRACES central North Atlantic (GA03, between station 12 and 20). The data show significantly higher Ra-228 concentrations (2.23 ± 0.41 dpm/100L) than the values observed in the central South Atlantic (< 0.3 dpm/100L; Hanfland, 2002, ANT XV/4). Moore et al. (2008) have found a similar distribution of Ra-228 between the North and South Atlantic. This value is more than 2/3 of the highest value (3.22 dpm/100L) observed in the shelf water of the Cape Basin in this study. Therefore, this is not a suitable background value for the South Atlantic waters. In contrast, the mid-water data from GA03 (1000-3000 m depth, between station 12 and 20, and 1000 m above seafloor) show a very similar background value (0.16 ± 0.1 dpm/100L) to this study. Thus, we prefer to use the low background value to reflect the Ra-228 level in the South Atlantic. In the revised manuscript, we have added the comparison between the data in the North and South Atlantic (Line 206-210).

Line 185: This sentence states that the two scenarios (mixing only, diffusive only) are used to bracket the range of estimates. However, it is stated in line 193 that the diffusive only case is used nearshore, and the mixing only case is used past the shelf-break. Throughout the methods section, it is confusing which assumptions are applied in which environments (e.g. when vertical mixing is ignored, or when advection is ignored)- perhaps it would help to structure the methods section by geographic region rather than by the individual assumptions? I am not entirely convinced that an advection-only scenario is valid. In other words, what is the basis for ignoring diffusive mixing? Are there other studies that have used Ra-228 in this way before? To the best of my knowledge, most other Ra-based studies ignore advection rather than diffusion.

***We agree with the reviewer. The advection-only scenario is unlikely to be valid. We only do this as an example for comparisons. In this study, it has only been applied to the data in the Argentine Basin after the shelf-break where the advection signal is strong. We have rewritten the relevant section to clarify that in the revised manuscript (Line 217-219).

Results:
Line 214-215: delete "with expectations based on" so the sentence reads "are consistent with GEOSECS and TTO observations. . ."

***We have made the change (Line 270-272).

Line 213: Are the authors specifically comparing the vertical profiles with Southern Atlantic GEOSECS and TTO stations here, or the entire Atlantic? If the entire Atlantic, the Charette et al (2015) North Atlantic GEOTRACES data should also be referenced.

***The comparison is only for the South Atlantic. We have clarified this in the revised manuscript (Line 284-286).

Discussion: Vertical mixing:
Does the exponential curve fit for Ra include any of the data in the surface layer, or does the curve fit start below the mixed layer (Figure 6)?

***The exponential curve fit starts from below the mixed layer but includes the average value in the mixed layer at the bottom of the layer. We have explained this with more details in the revised figure caption (Line 840-843).

Station 2 does not have sufficient Ra data (only one point) below the mixed layer to fit a curve. This station should be removed from the vertical mixing analysis.

***We agree with the reviewer that Station 2 does not have sufficient Ra data below the mixed layer. However, the mixing result is consistent with the adjacent stations in the Cape Basin. Therefore, we prefer to keep the mixing results for comparison in Fig. 6, but we have removed the Stn2 vertical TE flux calculations, and hence the Stn2 TE data from Fig. 4 and Table B2. We have also explained the limitation due to the insufficient Ra data in the revised manuscript (Line 347-349).

Some of the linear regressions on the trace element vertical profiles start below the mixed layer, while others do not (Figure 4). Why does this vary from station to station? In particular, the Zn data has particularly poor linear fits, and it is not clear whether all the data were included in the fit (the deepest sample at all three stations are below the end of the dashed line).

***As mentioned in the beginning, we used the function for linear regression in Excel to fit all the available TE data in the top 600 m. By default, this fit mistakenly assumed that the TE concentration (x-axis) is the dependent variable and that the depth (y-axis) is the independent variable, but it should be the other way round. This issue has now been corrected. Most of the gradients stay the same (except for Zn, slightly steeper), and hence the changes in the TE fluxes are insignificant. We have updated the figures (Fig. 3 and 8) and results (Table 3 and 4) in the revised manuscript.

Box model/trace element inputs:
Why are the calculations of Ra/TE fluxes put in an appendix? I recommend moving these into the main text, as they form the basis for the conclusions of the paper.

***We agree with the reviewer that the TE flux calculations form the basis for the conclusions of the paper, but think that they would also obstruct the flow of discussion in their current form. Thus, we have moved the basics of the calculations into the main text by adding a new section in the methods (2.5) to introduce these approaches and equations. We have kept the calculation details in the appendix to avoid any potential obstruction in the discussion.

Line 305: change "may not be unreasonable" to "are reasonable"

***We have made the change (Line 385-387).

Lines 309 – 311 are repetitive with line 299-301

***We have made the change (Line 380-383).

Line 319: delete "in fact" so the sentence reads "which is very close.."

***We have deleted this (Line 397-399).

Line 321: I recommend saying that the Fe fluxes are "slightly lower than" previously reported fluxes, instead of saying that they "compare well with" these other estimates, as they are an order of magnitude lower.

***We have made the change (Line 400-402).

Line 328: I don't understand why these estimates must represent a lower bound for what is expected in the South Atlantic. If the concentrations of Zn are lower in the Southern Atlantic, wouldn't a lower flux be expected, making these reasonable estimates rather than lower bounds?

***We have rewritten the sentence in the revised manuscript (Line 407-409).

Line 335-340: The Southern Ocean Fe fluxes reported by Dulaiova et al (2009) are almost exactly equal to the winter mixing Fe fluxes in the North Atlantic reported by Achterberg et al (2018). However, the authors state that the Fe fluxes they calculated are similar to Dulaiova but lower than Achterberg.

***We have corrected the statement in the revised manuscript (Line 419-421).

Line 363: stating that the integrated timescale is "relatively short" makes it seem like the half life of Th-234 is too short to capture changes in POC flux. It could be the opposite, that the biology is changing faster than the Th. I suggest removing "the integrated timescale is relatively short" so the sentence reads "given that the mean life of Th is 35 days". This way both possible conditions are accounted for.

***We have made the change (Line 447-450).

The loss via sinking out of the surface box is estimated using 234Th-based POC fluxes, and TE/C uptake ratios in phytoplankton. As I understand it, using the TE/C ratios in phytoplankton aims to isolate the flux from biology. Why not estimate the total flux? The box model sources include inorganic inputs, so the total source from sinking particles should be considered, not just the organic flux. This can be done by multiplying TE/234Th ratios by Th fluxes, rather than adding the extra step of converting to C in the middle. TE/234Th ratios are available for the North Atlantic (Hayes et al., 2018) and South Pacific (Black et al., 2019). If it is not possible to find TE/234Th ratios for all metals, perhaps this can at least serve as a comparison to the biology-based fluxes for the metals where data is available.
In Hayes et al. (2018), the total Fe flux is significantly smaller than the biological flux estimated here; perhaps the lab-based Fe/C ratios over-estimate the actual ratios in this region?

***We agree with the reviewer that the total TE fluxes (biological uptake + particle scavenging) are more appropriate to represent the total output. However, it is not

unreasonable to assume that biological uptake is the major output for the chosen trace elements in the open ocean primary productivity zone. Particle scavenging may become more important in certain areas of the surface ocean (e.g. continental margins or high dust plume regions), but fluxes of micronutrients metal in the region in this study is likely to be dominated by biological uptake.

Although the direct comparison of TE/234Th ratios between this study and Hayes et al. (2018) or Black et al. (2019) is currently not available, the comparison of the pFe export fluxes between this study and Hayes et al. (2018) actually shows relatively good agreement – 31 ~ 970 nmol/m2/day (this study) vs 274 ~ 2740 nmol/m2/day (0.1 ~ 1 mmol/m2/yr, the results of the 234Th approach between the longitude 24.5W and 66.5W shown in the Fig. 8c in Hayes et al., 2018). Likewise, the estimates of pCo flux between this study and Hayes et al. 2018 show similar level results (1 ~ 29 nmol/m2/day vs 0.27 ~ 6.8 nmol/m2/day, converted from 0.1 ~ 2.5 umol/m2/yr). We have added this comparison in the revised manuscript (Line 442-444).

The depth horizon for the surface box is not mentioned in the text, but is noted as 50 m in Figure 8. Why was this depth chosen? Please add an explanation to the text in addition to noting the depth in the figure. Particle fluxes can vary greatly depending on the depth chosen- see Buesseler et al. 2020 for example.

***The depth of 50 m was initially chosen as an average depth of the mixed layer in this region, and it was only used (to get the vertical section area) in the calculations of the net TE fluxes from the continental shelf to the defined surface box in the open ocean. We agree with the reviewer that the 234Th particle fluxes can vary with the integration depths. The 234Th flux is usually integrated to the 234Th-238U equilibrium depth (e.g. Thomalla et al., 2006) or to the 1% light depth (e.g. Owens et al., 2015). Thus, we compare the 234Th POC fluxes around the 40S Atlantic from Thomalla et al. (2006) (7.0 ± 2.2 mmol/m2/day, CTD05, integration depth: 101 m) and Owens et al. (2015) (6.4 ± 3.3 mmol/m2/day, DT6, integration depth: 88 m). Despite the slight difference in the integration depths, the two independently assessed POC fluxes are relatively consistent in this region.

To reflect the integration depth of the removal flux in the euphotic zone in the box model, we have changed the depth of the surface box to 100 m and corrected the shelf-ocean TE fluxes accordingly. We have provided more details in the revised manuscript (Line 434-442).

In the conclusion (line 383) and abstract the authors mention that particle inputs may need to be accounted for to close gaps in the mass balance. What are some examples of these possible particle sources? Is dust deposition not considered a particle source? It seems like some particle inputs and outputs are considered in the box model but not others.

***The particles here mainly refer to the laterally transported particles from the continental margins. Rijkenberg et al. (2014) have suggested that laterally transported particles from the offshore export waters may release Fe to the upper ocean in this region. However, it is not possible to evaluate this flux with our current dataset. We have clarified this in the revised manuscript (Line 456-460).

Figures:

Figure 1a: It would be helpful to add "Cape Basin" and "Argentine Basin" to the map, as these locations are referenced throughout the text.

***We have added the ocean basin names to the revised map.

Figure 1a: "GEOSEC" should be changed to "GEOSECS"

***We have made the correction.

Figure 1b: The colorbar extends over a much larger range than necessary; as far as I can tell, there are no samples with a salinity below 34. Shorten the range of the colorbar (e.g. 34 – 36) so that changes in the study area can be more clearly seen. I also recommend using a different color scale- this one has non-linear changes in both hue and brightness that make some gradients appear sharper than others (and will not print well as black & white). The highest peach values can also be misconstrued as orange (falling lower on the color scale between yellow & red, as opposed to being read as the highest values).

***We have made the change in the revised manuscript.

Figure 4: The caption says that dashed lines show linear regression trends with the uncertainty. It's not clear how the uncertainty is shown- should there be shading around the line? Or are the authors referring to the equations next to the lines?

***The uncertainty refers to the slope in the equation. We have clarified this in the revised caption.

Figure 4: Why do some of the lines not extend through all of the data? (e.g. panels g and j)

***As mentioned above, the Excel linear regression was performed mistakenly assuming that the TE concentration (x-axis) is the dependent variable and the depth (y-axis) is the independent variable, when it should be the other way round. This has been corrected in the revised manuscript.

Tables:
Table 1: On my PDF, it appears as if half the caption is above the table and half is below. I am not sure if the authors intended to put some information at the end (information about extrapolated Ra-226 activities & errors), but this should be included in the top caption.

***The information at the end of the table is the footnotes. We have labelled and clarified this.

AppendixD:
Line 400: The end of this sentence is missing.

***There should be a colon symbol in the end of this sentence. This sentence has been removed from the revised manuscript.

Line 444: What two surfaces are the authors referring to?

***In Fig 8, the two surfaces refer to the surface of the shelf horizontal plane (brown) and the surface of the shelf vertical section (red). We have clarified this in the revised manuscript (Line 512-514).

References:
Buesseler et al. 2020. "Metrics that matter for assessing the ocean biological carbon pump." Proceedings of the National Academy of Sciences. DOI: 10.1073/pnas.1918114117
Hayes et al. 2018. "Flux of Particulate Elements in the North Atlantic Ocean Constrained by Multiple Radionuclides." Global Biogeochemical Cycles. https://doi.org/10.1029/2018GB005994
Black et al. 2019. "Insights From the 238U-234Th Method Into the Coupling of Bio- logical Export and the Cycling of Cadmium, Cobalt, and Manganese in the Southeast Pacific Ocean" Global Biogeochemical Cycles. https://doi.org/10.1029/2018GB005985

**Reviewer #2**

General Comments The scientific quality of this paper is very good and the quantitative results seem both robust and oceanographically consistent. The calculations and argument are concise and the purpose of the paper is very clear. The comparison of different methods used for radium calculations is useful and provides confidence in the results. However, it seems like the radium side of the paper is significantly more developed than the trace metal side. A little more context and detail could be given to the scientific significance of calculating trace element inputs, and more discussion given to the implications of the calculated trace metal fluxes. Also, the methods of the trace metal calculations, which are an integral part of the paper and several sentences of the abstract, could be moved from the appendix to the main body of the paper. One calculation, the Co/C, Fe/C, and Zn/C export fluxes are noted for being significantly higher than corresponding dissolved inputs. This could be due to the use of laboratory culture metal quota values that were grown under replete conditions. As noted in the paper this region and section has some of the lowest Zn values observed in the euphotic zone from the prior study. Moreover, phytoplankton (and microbes in general) have extraordinary plasticity with respect to metal content, and the lower field and lab scarcity cultures are likely considerably more representative. It would be useful to redo these calculations with metal-carbon values that are more realistic, either using culture studies of phytoplankton grown under scarce rather than replete conditions (e.g. from the relevant Sunda and Huntsman studies) or using particulate metal data from other field expeditions if that data is not available from this section. This would probably help resolve the imbalance in the fluxes compared to the other fluxes calculated and would be more correct in construction of the calculations.

***We thank the reviewer for their positive feedback and constructive comments, particularly for the discussion of trace elements. We respond to all the comments point by point below and explain how we have addressed the issues in the revised manuscript. Following the reviewer's suggestion, we have also applied a range of reasonable estimates

for the TE/C ratios (based on Sunda and Huntsman studies) to recalculate the TE removal fluxes in this region (Line 433-444).

Specific Comments Introduction: Why are we interested in a calculation of Co, Zn, and Fe flux? Why those elements and not others? Please give more context for trace metals in this region, and the significance of the calculations presented here.

***Iron, zinc and cobalt are known to be essential micronutrients for marine phytoplankton. They all play important roles in the cellular metabolic enzymes and sometimes can collimate primary productivity in different ocean regions. Previous studies have shown that the study region near the 40oS Atlantic is mainly iron limited (Moore et al., 2004; Browning et al., 2014; 2017) and has the lowest reported dissolved zinc concentrations in the global ocean (Wyatt et al., 2014). The requirement for Zn in phytoplankton can be replaced by Co in low Zn areas (Price and Morel, 1990). Thus, knowing the sources and fluxes of these three elements can improve our understanding of the limiting factors for productivity in this study region, one of the most dynamic nutrient regimes in the oceans. We have followed the reviewer's suggestion and added more context in the introduction to highlight the importance of these three elements in this region (Line 33-43).

Line 40: Define which metals you'll be talking about (Fe, Zn, Co) here, or at least within the first few paragraphs of the introduction, and why they're important in this region.

***As mentioned above, we have clarified this and added more context in the introduction to highlight the importance of these elements in this region (Line 33-43).

Lines 83-91: This paragraph should be moved out of the methods section to the intro- duction or to the results, perhaps as a "hydrographic setting" section in the beginning of the results.

***We agree with the reviewer, but we feel that this paragraph does not fit in either the introduction or the results section. To address this issue, we have thus changed the methods section title to "2 Study sites and methods". We have then split the relevant subsection into two, "2.1 Hydrographic setting" and "2.2 Water sampling".

Line 94, 104: Define your trace metal clean technique in more detail. What acids were used and for how long?

***All the trace-metal cleaning procedures follow the GEOTRACES sampling protocols (Cutter et al., 2010). The sample tubing and bottles were rinsed with Milli-Q water and filled with 0.1M HCl for one day. After emptying the acid, the tubing and bottles were rinsed thoroughly with Milli-Q water. The tubing and bottles were also rinsed with open-ocean seawater before sampling. We have added more details in the revised manuscript (Line 117-120).

Lines 142-5: Useful analysis of the two methods' uncertainty and error differences. I suggest also performing a pairwise t-test to compare the 226Ra data generated from direct observation and from the Si estimation to determine if the two methods are statistically similar, even with the larger error on the estimated method. This would help convince readers that the methods are comparable.

***Follow the reviewer's suggestion, we have performed a paired t-test and the p-value is 0.55 (> 0.05), which confirms that the measured Ra-226 and the Si-extrapolated Ra-226 data are not statistically significantly different. We have added the t-test results in the revised manuscript (Fig. A1 and Line 158-160).

Line 195: Include sections in the methods to clearly describe how Kz was calculated and the vertical flux of trace metals. They are critical results in your paper and are given prominence in your title and abstract, and their methods should not be buried in your appendix. Perhaps an abbreviated description of the appendix D calculations can be described in the methods, and the full version can be in the appendix.

***We have added more details to describe the calculation of Kz in Section 2.4 (Line 229-236). As discussed in the response to Reviewer 1 comments, we have summarised the basics of the flux calculations in the main text by adding a new section in the methods (2.5) and kept the calculation details in the appendix.

Lines 211-4: Make this point clearer. You imply but don't clearly state that your results are more similar to Atlantic values than Southern Ocean values. That is an interesting result in this transition zone and should be made more explicit.

***We have rewritten these sentences to clarify this (Line 282-284).

Lines 215-22: Expand more on this entire section and convey a more detailed picture of TE distributions in this region. Even if the data has been described elsewhere, this section is too short. Additionally, both Fe and Co do not typically continue to "increase with depth below the mixed layer" because they're scavenged-type elements. Fig. 4 implies that all 3 trace elements increase linearly with depth, which is not necessarily the case beyond the mesopelagic. Give more context to the distribution of all three trace metals here, and qualify that the TEs (specifically Fe and Co) only increase with depth in the upper ocean.

***We have added more context to describe the distribution of these three trace elements in the study region (Line 290-302). We have also clarified that the increase in TE concentrations with depth only applies to the upper ocean (Line 299-300).

Lines 261-7: Move the description of the Kz calculation to the methods and include the explicit equation for its calculation.

***We have moved the Kz calculation to the method section (2.4) and added more details to describe the calculation (Line 229-236).

Line 298: As already stated, move at least a brief description of the calculation to the methods section.

***As mentioned above, we have summarised the basics of the flux calculations in the method section (2.5).

Line 317: Briefly expand on low oxygen resulting in higher Co fluxes and provide a citation.

***We have added a few sentences to explain the relationship between oxygen and cobalt fluxes with relevant references (Line 395-397).

Lines 322-5: Discuss your findings more than just comparing them to other fluxes. Why do you think they are lower than the other reported Fe fluxes you cite?

***As already mentioned in the manuscript, these high-Fe fluxes are particularly found in regions with river plumes, SGD and the oxygen minimum zone. We have clarified this in the revised manuscript (Line 402-405).

Lines 327-30: Same as the above comment for Fe. Why do you think this region has a lower Zn flux? Was this expected?

***There is very limited oceanic Zn flux available for comparison. The only available estimate from the western North Atlantic (Charette et al., 2016) indicates that the cross-shelf Zn fluxes in the South Atlantic are lower than the western North Atlantic. The results may explain why the South Atlantic has the lowest reported Zn concentration in the global oceans. However, it still needs more detailed studies to understand the reasoning behind the different Zn fluxes, which is beyond the scope of this study. We have highlighted this in the revised manuscript (Line 409-410).

Lines 341-3: This paragraph seems out of place. Either expand on why you're reporting aerosol data and give context for the fluxes (how they compare to the vertical TE fluxes, etc.) or simply move these two sentences to section 4.4 where the numbers are used.

***We have moved this paragraph to the beginning of the section (Line 373-376), and emphasised that apart from the atmospheric dust deposition, other inputs are still poorly constrained in this region.

Lines 361-72: Specify which of these 3 hypotheses are supported by your results. It sounds like the particles might be a good answer, but it's only mentioned briefly. Expand a bit on what you think is likely going on.

***As discussed in the response to Reviewer 1 comments, the particle hypothesis is based on the observation from Rijkenberg et al. (2014). They have observed high dissolved Fe in the mid depths of this region and suggested that the laterally transported particles from the offshore export waters may release Fe to the upper ocean. However, it is not possible to provide more discussion without further details of the sources and TE concentrations of these particles and the mechanism releasing TEs from these particles. We have highlighted this in the revised manuscript (Line 456-460).

Figure 3 (lines 745-750) The correlations of metals with salinity is very interesting and worth emphasizing a bit more. Similar observations have been observed on the North American shelf as well observed by Bruland and Franks 1984 and Noble et al. 2017. It could be

worthwhile to point out that these observations suggest this is a general feature of Atlantic Western boundaries.

***These observations imply that the distribution of trace elements in the surface ocean is strongly associated with the shelf-water masses in the western boundaries of the Atlantic. We have added a few sentences to highlight this in the revised manuscript (Line 297-299).

Bruland, K. W., & Franks, R. P. (1983). Mn, Ni, Cu, Zn and Cd in the western North Atlantic. In Trace metals in sea water (pp. 395-414). Springer, Boston, MA.
Noble, A. E., Ohnemus, D. C., Hawco, N. J., Lam, P. J., & Saito, M. A. (2017). Coastal sources, sinks and strong organic complexation of dissolved cobalt within the US North Atlantic GEOTRACES transect GA03. Biogeosciences, 14(11).

Technical Comments
General: Keep formatting choices consistent. Both 1-D and 1D are used to describe the model, and spaces are inconsistently placed between a number value and its unit. (ex. line 140 uses 100L-1 and line 141 uses 100 L-1.)

***We have checked the formatting and ensured consistency in the revised manuscript.

General: Are significant figures to tens of pM for dCo correct?

***This is correct.

Line 102: Define RaDeCC here. It is defined later on line 111, and should be defined at the first use of the abbreviation.

***We have defined this in the revised manuscript (Line 122).

Line 114: Define MC-ICP-MS here.

***We have defined this in the revised manuscript (Line 125).

Lines 116-7: Briefly clarify why it is necessary to convert from Sr(Ra)SO4 to Sr(Ra)CO3.

***As sulfate is much harder to redissolve back in solutions, the conversion from sulfate to carbonate can improve the efficiency of dissolution. We have clarified this in the revised manuscript (Line 128-130).

Line 119: Briefly describe what AG50-X8 and Sr-Spec are. You describe them as ion exchange columns on line 122, but that should be made clearer when they're first mentioned.

***We have clarified these in the revised manuscript (Line 130-133).

Lines 126-7: This sentence was already stated on line 114, but without mention of 226Ra. Are you restating the analyses used, or is this a different method? It is not clear.

***This sentence is redundant. We have rewritten and combined it with the next sentence in the revised manuscript (Line 141-143).

Line 127: Change Ra-228 to 228Ra to be consistent with the rest of the paper.

***As mentioned above, this sentence has been rewritten and therefore it should not be a problem now. However, if an isotope is used in the beginning of a sentence, conventionally the element goes first to avoid beginning the sentence with a number.

Lines 138-40: This is a run on sentence despite the use of parentheses. I suggest removing the parentheses and forming two separate sentences.

***We have rewritten the sentence (Line 153-155).

Line 199: "show" not "shows".

***We have corrected this (Line 271).

Line 253: Use a period or semicolon between "waters" and "therefore" to avoid the run on sentence.

***We have made the change (Line 335).

Line 283: Figure 7 only shows depth, not sigma-t as this line states. Rephrase this sentence or relabel the figure to show the density ranges instead of depths.

***We have rephrased the sentence (Line 362-363).

Line 297: Is the shelf 228fluxes point a fourth part of this list, or is it a separate point from the 3 approaches list? If it's part of the list, label it as (4). If it's a separate point, put it in its own sentence.

***The shelf 228Ra flux is only used for the TE/228Ra-ratio-derived TE fluxes. The sentence has been rewritten to clarify this in the revised manuscript (Line 377-379).

Lines 309-10: Confusing sentence – nothing is being compared here. Rephrase.

***The sentence has been rephrased to avoid confusion (Line 391-395).

Figures: The figures here do a good job of supporting the data and conclusion of the work, but they should be edited for readability. In general, do not rely on color coded axes to label subplots (especially for colorblind people), and include axis labels with units clearly next to the axis number scale. Avoid breaking up the axis label and the axis unit. Also, consider putting some of the regression equations in a table or in the appendix, particularly if they're not referenced in the text.

\*\*\*We have checked and corrected these problems in the revised manuscript. We have also simplified the regression equations to the key components that are relevant to the equations referenced in the text for the mixing and flux calculations (Fig. 3-6).

Fig 1: The stations numbers are crammed into the top plot b and are mostly illegible. The Salinity z-axis label is also in a strange location. I suggest leaving more space between plots a and b to include this information above figure b.

\*\*\*We have made the changes.

Fig 4: Add y-axis labels for depth. Also, the x-axis labels are difficult to find. Is there a way to make the metal label clearer for each row?

\*\*\*We have added the depth label and clarified the metal label for each row.

Fig 6: The x-axis labels for the top row of plots are not very readable here. The 228Ra label should be next to its units and ideally should be above the axis like the labels for the bottom row of plots. The subtle color coding for sigma-t and T is also confusing.

\*\*\*We have remade the figure to avoid these problems.

\*\*\*Other references cited in the response:

Browning et al. (2014) Nutrient regimes control phytoplankton ecophysiology in the South Atlantic, Biogeosciences, 11, 463-479.

Browning et al. (2017) Iron limitation of microbial phosphorus acquisition in the tropical North Atlantic, Nature Communications, 8, 15465.

Charette et al. (2016) Coastal ocean and shelf-sea biogeochemical cycling of trace elements and isotopes: lessons learned from GEOTRACES, Phil. Trans. R. Soc. A, 374, 20160076.

Clough et al. (2016) Measurement uncertainty associated with shipboard sample collection and filtration for the determination of the concentration of iron in seawater, Anal. Methods, 8, 6711-6719.

Cutter et al. (2010) Sampling and sample-handling protocols for GEOTRACES cruises. http://www.geotraces.org/libraries/documents/Intercalibration/Cookbook.pdf

Geibert et al. (2013) 226Ra determination via the rate of 222Rn ingrowth with the Radium Delayed Coincidence Counter (RaDeCC), Limnology and Oceanography Methods, 11, 594-603.

Hanfland (2002) Radium-226 and Radium-228 in the Atlantic sector of the Southern Ocean, PhD thesis, AWI Bremerhaven Germany, 135pp.

Hsieh and Henderson (2011) Precise measurement of 228Ra/226Ra ratios and Ra concentrations in seawater samples by multi-collector ICP mass spectrometry, Journal of Analytical Atomic Spectrometry, 26, 1338-1346.

Moore et al. (2004) Upper ocean ecosystem dynamics and iron cycling in a global three-dimensional model. Global Biogeochemical Cycles, 18, GB4028.

Moore et al. (2008) Submarine groundwater discharge revealed by 228Ra distribution in the upper Atlantic Ocean, Nature Geosciences, 1, 309-311.

Owens et al. (2015) Thorium-234 as a tracer of particle dynamics and upper ocean export in the Atlantic Ocean, Deep Sea Research, Part II, 116, 42-59.

Price and More (1990) Cadmium and cobalt substitution for zinc in a marine diatom, Nature, 344, 658-660.

Rijkenberg et al. (2014) The distribution of dissolved iron in the west Atlantic Ocean, PLoS One, 9, e101323.

Thomalla et al. (2006) Particulate organic carbon export from the North and South Atlantic gyres: the 234Th/238U disequilibrium approach, Deep Sea Research Part II: Topical Studies in Oceanography, 53, 1629-1648.

Wyatt et al. (2014) Biogeochemical cycling of dissolved zinc along the GEOTRACES South Atlantic transect GA10 at 40°S, Global Biogeochem. Cy., 28, 44-56.

Wyatt et al. (2020) Seasonal cycling of zinc and cobalt in the Southeast Atlantic along the GEOTRACES GA10 section, Biogeosciences Discuss., in review.

---

## Author Response (AR2)

**Response to Editor's and Reviewer's comments on "Radium-228-derived ocean mixing and trace element inputs in the South Atlantic" by Hsieh et al. (bg-2020-377)**

**Associate Editor**
Comments to the Author:
Dear Yu-Te et al.,

Once you have changed the typo in the introduction, I'll be happy to accept your manuscript for publication.

Sincerely,
Markus

**Reviewer #1**
There is a typo in the first paragraph of the revised introduction: on line 41 of the revised manuscript, "practically" should be "particularly".

Dear Editor and Reviewer,
Thank you for your time and thorough review. We have corrected the typo in the revised manuscript. We have also updated a reference (Wyatt et al., 2021) which has been accepted for publication recently. We hope that the revised manuscript is now acceptable for publication in *Biogeosciences*.

Yours sincerely,

Yu-Te Alan Hsieh (on behalf of all co-authors)